# Intrinsically regulated learning is modulated by synaptic dopamine signaling

Pablo Ripollés[1,2,3†]*, Laura Ferreri[1,2†], Ernest Mas-Herrero[4,5,6], Helena Alicart[1], Alba Gómez-Andrés[1], Josep Marco-Pallares[1,2], Rosa Maria Antonijoan[7,8], Toemme Noesselt[9,10,11], Marta Valle[7,12†], Jordi Riba[13†‡], Antoni Rodriguez-Fornells[1,2,14†]*

[1]Cognition and Brain Plasticity Group, Bellvitge Biomedical Research Institute-IDIBELL, L'Hospitalet de Llobregat, Barcelona, Spain; [2]Department of Cognition, Development and Educational Psychology, Campus Bellvitge, University of Barcelona, L'Hospitalet de Llobregat, Barcelona, Spain; [3]Department of Psychology, New York University, New York, United States; [4]Montreal Neurological Institute, McGill University, Montreal, Canada; [5]International Laboratory for Brain, Music, and Sound Research, MontrealQC, Canada; [6]Centre for Research on Brain, Language and Music, Montreal, Canada; [7]Department of Pharmacology and Therapeutics, Universitat Autònoma de Barcelona, Barcelona, Spain; [8]Centre d'Investigació de Medicaments, Servei de Farmacologia Clínica, Hospital de la Santa Creu i Sant Pau, Barcelona, Spain; [9]Department of Neurology, Otto-von-Guericke University, Leipziger Straße, Magdeburg, Germany; [10]Department of Biological Psychology, Otto-von-Guericke-University Magdeburg, Postfach, Magdeburg, Germany; [11]Center for Behavioral Brain Sciences, Magdeburg, Germany; [12]Pharmacokinetic/Pharmacodynamic Modeling and Simulation Group, Sant Pau Institute of Biomedical Research, Barcelona, Spain; [13]Human Neuropsychopharmacology Group, Sant Pau Institute of Biomedical Research, Barcelona, Spain; [14]Catalan Institution for Research and Advanced Studies, Barcelona, Spain

*For correspondence:
pabloripollesvidal@gmail.com (PR);
antoni.rodriguez@icrea.cat (AR-F)

†These authors contributed equally to this work

Present address: ‡Department of Neuropsychology and Psychopharmacology, Maastricht University, Maastricht, The Netherlands

Competing interests: The authors declare that no competing interests exist.

**Abstract** We recently provided evidence that an intrinsic reward-related signal—triggered by successful learning in absence of *any external feedback*—modulated the entrance of new information into long-term memory via the activation of the dopaminergic midbrain, hippocampus, and ventral striatum (the SN/VTA-Hippocampal loop; Ripollés et al., 2016). Here, we used a double-blind, within-subject randomized pharmacological intervention to test whether this learning process is indeed dopamine-dependent. A group of healthy individuals completed three behavioral sessions of a language-learning task after the intake of different pharmacological treatments: a dopaminergic precursor, a dopamine receptor antagonist or a placebo. Results show that the pharmacological intervention modulated behavioral measures of both learning and pleasantness, inducing memory benefits after 24 hr only for those participants with a high sensitivity to reward. These results provide causal evidence for a dopamine-dependent mechanism instrumental in intrinsically regulated learning and further suggest that subject-specific reward sensitivity drastically alters learning success.
DOI: https://doi.org/10.7554/eLife.38113.001

## Introduction

Growing evidence both from animal and human studies support the notion that midbrain dopaminergic neurons of the substantia nigra/ventral tegmental area complex (SN/VTA), along with the ventral striatum (VS) and the hippocampus (HP), form a functional loop (the SN/VTA-HP loop) in the service of learning and memory (*Lisman and Grace, 2005*; *Goto and Grace, 2005*; *Lisman et al., 2011*; *Shohamy and Adcock, 2010*; *Kamiński et al., 2018*). In the downward arm of the circuit, signals are sent from the HP to the SN/VTA through the VS, which is thought to integrate affective, motivational, and goal-directed information into the loop (*Lisman and Grace, 2005*; *Goto and Grace, 2005*). In the upward arm of the loop, dopamine is released from the SN/VTA back into the HP, which in turn enhances memory formation and learning through long-term potentiation (LTP) processes (*Lisman et al., 2011*; *Lisman and Grace, 2005*; *Shohamy and Adcock, 2010*). Within this loop, dopamine plays a critical role, as its release promotes the creation of stable memories by allowing LTP to persist over time (*Bethus et al., 2010*; *Frey et al., 1990*; *Hansen and Manahan-Vaughan, 2014*; *Huang and Kandel, 1995*; *McNamara et al., 2014*; *Rossato et al., 2009*).

In this vein, fMRI research in humans has consistently shown that both explicit (*Adcock et al., 2006*; *Wittmann et al., 2005*; *Wolosin et al., 2012*; *Callan and Schweighofer, 2008*) and implicit *reward* (*Ripollés et al., 2016*), and even intrinsic motivational states (i.e., curiosity; *Gruber et al., 2014*), can promote the storage of new information into long-term memory through the activation of the SN/VTA-HP loop (see Figure 8 in, *Ripollés et al., 2016*). However, although fMRI activity within the SN/VTA is usually associated with the release of dopamine (*Düzel et al., 2009* ; *Ferenczi et al., 2016*; *Knutson and Gibbs, 2007*; *Salimpoor et al., 2011*; *Schott et al., 2008*), neuroimaging studies can only provide *indirect* evidence of the actual involvement of the dopaminergic mesolimbic system. In order to prove that a dopamine-dependent mechanism plays a critical role in learning and memory processes, one avenue to pursue is to directly manipulate dopaminergic neurotransmission in the human brain through pharmacological interventions. In this vein, several studies have shown that administration of dexamphetamine and methylphenidate (which increase dopamine concentrations in the synapse by blocking its reuptake; *Breitenstein et al., 2004*; *Whiting et al., 2007*; *Whiting et al., 2008*; *Linssen et al., 2014*) and specially, levodopa (the immediate precursor of dopamine) can enhance memory and learning in both healthy (*Shellshear et al., 2015*; *Bunzeck et al., 2014*; *Chowdhury et al., 2012*; *Knecht et al., 2004*) and clinical populations (*Berthier et al., 2011*).

We recently provided behavioral, functional and physiological evidence by means of fMRI and skin conductance response, showing that an intrinsic reward-related signal—triggered by successful learning in absence of *any external feedback or explicit reward*—modulated the entrance of new information into long-term memory via the activation of the SN/VTA-HP loop (*Ripollés et al., 2016*). Here, we used a double-blind, within-subject randomized pharmacological intervention to directly assess the hypothesis that synaptic dopamine availability plays a causal role in this learning process. A group of 29 individuals were asked to perform a language-learning task (that mimics our capacity to learn the meaning of new-words presented in verbal contexts; *Ripollés et al., 2014*; *Ripollés et al., 2016*; *Ripollés et al., 2017*; *Mestres-MisseMissé et al., 2007*) after the intake of three different pharmacological treatments: a dopaminergic precursor (levodopa, 100 mg +carbidopa, 25 mg), a dopamine antagonist (risperidone, 2 mg), or a placebo (lactose). Levodopa is rapidly taken up by dopaminergic neurons, transformed into dopamine and stored in vesicles from which it will be released into the synaptic cleft each time the neuron fires. Thus, levodopa leads to a general increase in dopamine available for release in brain areas innervated by dopaminergic afferents. On the other hand, risperidone—a dopamine antagonist—interferes with dopaminergic neurotransmission by binding with a group of receptors known as D2 or D2-like (*Burstein et al., 2005*). Therefore, in the presence of risperidone, the transmission of dopamine-mediated signals to post-synaptic neurons will be reduced due to the blockade of the D2 receptor family.

We aimed at assessing the influence of dopamine signaling on learning and reward using the pharmacological approach described above. Each of the two experimental sessions involving active drugs were intended to shift dopaminergic neurotransmission away from each individual's

physiological status, as measured in a placebo session, and in opposite directions: levodopa to enhance the dopamine availability for release into the synapse, and risperidone to reduce synaptic transmission of the dopamine-associated signal by hindering dopamine-receptor interactions (for the use of levodopa or risperidone during cognitive tasks, see e.g. *Rabella et al., 2016*; *Wittmann and D'Esposito, 2015*; *de Vries et al., 2010*; *Knecht et al., 2004*) . Accordingly, we predicted that behavioral measures of both learning and reward should increase and decrease under levodopa and risperidone, thus modulating—with opposite effects—the memory benefits for the learned words after a consolidation period (24 hr).

## Results

Twenty-nine healthy participants completed a behavioral version of our word-learning task (see Materials and methods), in which the meaning of a new-word could be learned from the context provided by two sentences built with an increasing degree of contextual constraint (*Mestres-Missé et al., 2010*). Only half of the pairs of sentences disambiguated multiple meanings, allowing the encoding of a congruent meaning of the new-word during its second presentation (M+ condition). For the other pairs, the new-word was not associated with a congruent meaning across the sentences and could not be learned (M- condition). This condition, as in our previous study (*Ripollés et al., 2016*), was included to control for possible confounds related to novelty, attention and task difficulty (*Guitart-Masip et al., 2010*; *Bunzeck and Düzel, 2006*; *Boehler et al., 2011*). At the end of each learning trial (i.e. after the second sentence for a particular new-word appeared), participants first provided a confidence rating (a subjective evaluation of their performance) and then rated their emotions with respect to arousal and pleasantness. After approximately 24 hr (no drug intake occurred during the second day of testing), participants completed a recognition test to assess their learning (chance level was 25%; see Materials and methods). Three participants were excluded from the analyses (see Materials and methods) and thus the final sample was reduced to 26 individuals (17 women, mean age = 22.27 ± 3.69).

We first assessed whether our participants' performance under the placebo condition replicated our previous results. Participants ascribed correct meaning to 60 ± 10% of new-words from the M+ condition during the encoding phase. In 61 ± 15% of the M- trials, participants correctly indicated an absence of coherent meaning. After 24 hr, participants still recognized the correct meaning of 65 ± 17% of learned new-words during the encoding phase [significantly above 25% chance level, $t(25)$ =12.28, p<0.001, d = 2.33; Bayes Factor-$BF_{10}$- equal to 1.9e + 9] and correctly indicated that 41 ± 22% of M- new-words identified during the encoding phase had no meaning ascribed [significantly above 25% chance level, $t(25)$=3.70, p<0.001, d = 0.70; $BF_{10}$ = 35.38].

In order to compare this performance with our previous results (24 participants from Exp. three in *Ripollés et al., 2016*), we submitted the learning scores to a mixed repeated measures ANOVA with Condition (M+,M-) as a within-subjects variable and Group (Pharmacological Group, Exp. 3 in *Ripollés et al., 2016*) as a between subjects variable. No significant effect of Group [Learning Day 1: $F(1,48)$=0.246, p=0.622, partial η2 = 0.005, $BF_{Inclusion}$ = 0.297; Recognition Day 2: $F(1,48)$=3.56, p=0.065, partial η2 = 0.069, $BF_{Inclusion}$ = 1.05] or Group × Condition interaction [Learning Day 1: $F(1,48)$=0.749, p=0.391, partial η2 = 0.015, $BF_{Inclusion}$ = 0.381; Recognition Day 2: $F(1,48)$=0.222, p=0.639, partial η2 = 0.005, $BF_{Inclusion}$ = 0.313] was found for the learning scores of Day 1 or the recognition rate after 24 hr. This shows that the new group of participants, during the placebo session, learned and remembered words from the M+ condition and correctly identified M- words (i.e. no meaning ascribed) at the same rate as in our previous experiment.

We then focused our analyses on learned (on Day 1) and still remembered (on Day 2) M+ new words. In our previous work (*Ripollés et al., 2016*), this was the condition associated to the largest fMRI activity within the SN/VTA-HP loop, the largest physiological response and the highest subjective pleasantness ratings, even when compared with learned words that were forgotten after 24 hr (as a control, we used M- new-words correctly identified during the encoding phase and after 24 hr). Accordingly, in the present study subjective pleasantness and confidence ratings on Day 1 were higher for remembered than for forgotten M+ new words in the 24 hr recognition test [pleasantness, $t(25)$=2.75, p=0.011, d = 0.42, $BF_{10}$ = 4.39; confidence, $t(25)$=4.56, p<0.001, d = 0.68, $BF_{10}$ = 232.82], while no difference in arousal ratings was encountered [$t(25)$=0.20, p=0.835, d = 0.025, $BF_{10}$ = 0.21]. In replicating our previous results (*Ripollés et al., 2016*), these findings

confirm that intrinsic reward (i.e. derived from an internal monitoring of learning success) had a modulatory effect on long-term memory. Regarding the M- control condition, as expected, there was no difference in subjective pleasantness [$t(25)$=1.40, p=0.172, d = 0.26, $BF_{10}$ = 0.49], arousal [$t(25)$ =1.28, p=0.212, d = 0.20, $BF_{10}$ = 0.43] or confidence ratings [$t(25)$=1.18, p=0.247, d = 0.18, $BF_{10}$ = 0.38] for M- new-words which were correctly identified during the encoding phase and still correctly rejected in the 24 hr test and those which were not. When submitting these ratings to a mixed repeated measures ANOVA with Condition (M+,M-) and Group (Pharmacological, Previous Data) as factors (in our previous analyses we excluded four participants from the rating analyses, see *Ripollés et al., 2016*; thus in this ANOVA we compare 20 participants from the previous dataset against 26 for the placebo session), no significant effect of Group [Pleasantness: $F(1,44)$=0.143, p=0.707, partial η2 = 0.003, $BF_{Inclusion}$ = 0.349; Arousal: $F(1,44)$=1.66, p=0.204, partial η2 = 0.036, $BF_{Inclusion}$ = 0.80; Confidence: $F(1,44)$=3.49, p=0.068, partial η2 = 0.073, $BF_{Inclusion}$ = 1.17] or Group × Condition interaction [Pleasantness: $F(1,44)$=0.239, p=0.627, partial η2 = 0.005, $BF_{Inclusion}$ = 0.321; Arousal: $F(1,44)$=0.216, p=0.645, partial η2 = 0.005, $BF_{Inclusion}$ = 0.304; Confidence: $F(1,44)$=0.028, p=0.868, partial η2 = 0.001, $BF_{Inclusion}$ = 0.280] was found. This shows that participants' ratings were also in line with those of our previous experiment (*Ripollés et al., 2016*).

Hence, we calculated the drug effect on the behavioral data. Specifically, for the levodopa and risperidone interventions and for each subject, we calculated the percentages of change in learning scores and behavioral ratings with respect to the placebo session (see Materials and methods). Notably, for the M+ condition, our findings show a pharmacological modulation of learning performance and behavioral reward ratings. The percentage of learned words during the encoding phase was higher under levodopa than under risperidone [as compared to placebo; $t(25)$=2.72, p=0.012, d = 0.56, $BF_{+0}$=8.26]. Importantly, this effect was still present at 24 hr for the total number of remembered new-words [$t(25)$=2.10, p=0.046, d = 0.45, $BF_{+0}$=2.62; see *Figure 1A*]. In addition, while no significant changes were found for the arousal ratings [$t(25)$=0.31, p=0.757, d = 0.049, $BF_{+0}$=0.26], the drug effect approached significance for confidence ratings [$t(25)$=2.05, p=0.051, d = 0.51, $BF_{+0}$=2.40] and was significant for pleasantness ratings [$t(25)$=2.70, p=0.012, d = 0.64, $BF_{+0}$=7.93], where scores for remembered words at 24 hr where higher under levodopa than under risperidone (as compared to placebo; see *Figure 1B*). There was not, however, a significant effect of drug on the *recognition rate* [i.e. the percentage of remembered words in the recognition test of Day 2 compared to those that were learned on Day 1; $t(25)$=-0.013, p=0.989, d = 0.003, $BF_{+0}$=0.20]. This suggests that the pharmacological intervention was able to modulate measures of reward, memory and online learning selectively during the main M+ condition (see the Individual Variability of the Drug Effect section of Appendix I, for a more in depth description of the individual differences found for the drug effect for each measure). Additional analyses using the values for the three sessions separately (instead of the percentage of change from placebo) further confirmed this pattern of results (see the Supplemental Behavioral Analyses section of Appendix 1 and *Figure 1—figure supplementary 1A–C*).

As expected, for the control M- condition no significant differences between the risperidone and levodopa interventions as compared to placebo were found for the online learning scores on Day 1 [$t(25)$=1.53, p=0.137, d = 0.28, $BF_{+0}$=1.08], the total number of correctly rejected M- words at 24 hr [$t(25)$=0.62, p=0.538, d = 0.15, $BF_{+0}$=0.35], the recognition rate [percentage of words correctly rejected on Day 2, in respect to those correctly rejected on Day1; $t(25)$=-0.04, p=0.968, d = 0.011, $BF_{+0}$=0.20; see *Figure 1A*], or the subjective ratings of arousal [$t(23)$=1.72, p=0.097, d = 0.45, $BF_{+0}$=1.46] and confidence [$t(23)$=0.36, p=0.720, d = 0.10, $BF_{+0}$=0.28; see *Figure 1B*; two participants were excluded from the rating analyses after not correctly rejecting any M- word at 24 hr from those correctly rejected during encoding in the levodopa intervention]. For the pleasantness ratings, however, the difference was close to significance [$t(23)$=2.02, p=0.055, d = 0.40, $BF_{+0}$=2.32]. However, it is important to note that the pleasantness ratings for M- trials remembered at 24 hr were *not different* from 0 at any session [Risperidone mean rating = −0.23, $t(23)$=-1.03, p=0.309, d = 0.20, $BF_{10}$ = 0.34; Placebo mean rating = 0.24, $t(23)$=1.01, p=0.319, d = 0.20, $BF_{10}$ = 0.34; Levodopa mean rating = 0.30, $t(23)$=1.20, p=0.240, d = 0.23, $BF_{10}$ = 0.40], implying that participants did not find this learning condition particularly rewarding even if the pharmacological intervention slightly modified their subjective ratings.

Given that our learning task modulates activity within the reward network and is associated with increased subjective reports of pleasure (*Ripollés et al., 2014*; *Ripollés et al., 2016*), we further

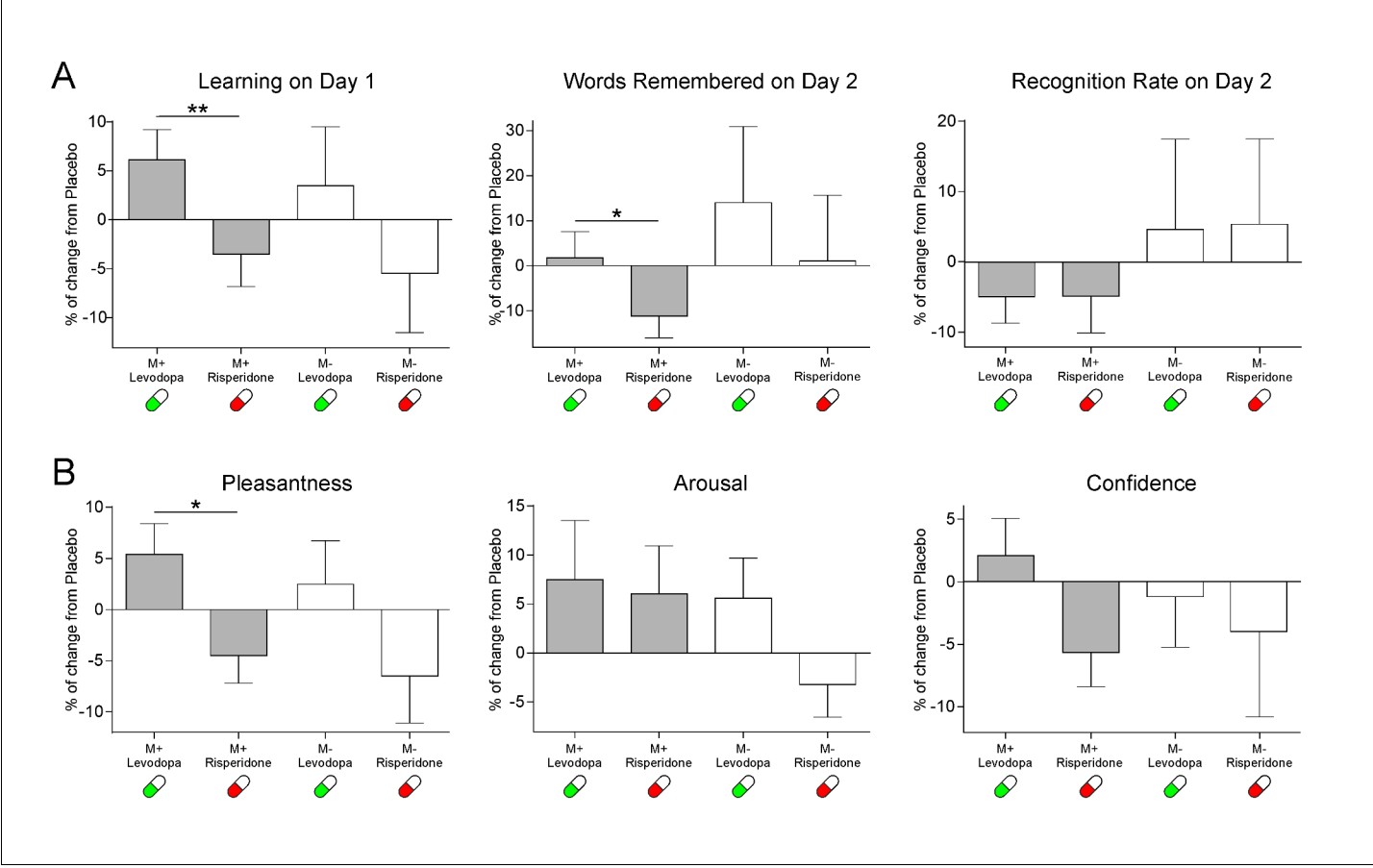

**Figure 1.** Effects of the pharmacological intervention (mean ± SEM) in (**a**) Learning and memory scores and (**b**) subjective ratings. Note that subjective ratings were only measured during the learning phase of Day 1. Effects are calculated as % of change with respect to the placebo session. *p<0.05, **p<0.01.

DOI: https://doi.org/10.7554/eLife.38113.002

The following figure supplement is available for figure 1:

**Figure supplement 1.** Mean plus standard error of the mean M+ scores for risperidone (black), placebo (white) and levodopa (grey) interventions separately for (**A**) learning and memory measures, (**B**) learning and memory measures separated for high (H+; black line) and low (H-; grey line) hedonic participants and (**C**) subjective ratings.

DOI: https://doi.org/10.7554/eLife.38113.003

tested whether individual differences in sensitivity to reward interacted with the drug intervention to modulate memory benefits (*Ferreri and Rodriguez-Fornells, 2017*; *de Vries et al., 2010*; *Apitz and Bunzeck, 2013*). Twenty-four out of the 26 participants completed the Physical Anhedonia Scale (PAS; *Chapman et al., 1976*); mean score = 11.62 ± 5.47). The PAS is a well-validated scale that evaluates difficulty in feeling physical and aesthetic pleasure in response to typical pleasurable stimuli (*Padrão et al., 2013*; *Mas-Herrero et al., 2014*). We correlated (using Spearman´s rho) participants' individual scores with the drug effect for each learning condition (the drug effect was calculated as the subtraction of the percentage of change from placebo of the levodopa intervention minus the percentage of change from placebo of the risperidone intervention, see Materials and methods). As a control and in order to take into account previous results (*Chowdhury et al., 2012*), we also assessed the relationship of the learning scores with the weight-dependent measure of drug dose (calculated in mg of levodopa/risperidone administered per kilogram, mean value = 1.66 ± 0.23). As expected, no significant correlations were found between the M- learning scores and the PAS [Learning Day 1 $r_s$ = −0.19, p=0.372; number of correctly rejected words during Day 2, $r_s$ = −0.34, p=0.097; recognition rate, $r_s$ = −0.19, p=0.372]. In addition, no significant linear correlation or inverted U-shape relationship (*Chowdhury et al., 2012*) was found for any learning

score (M+ or M-) and the weight dependent drug dosage (all ps > 0.13). However, for the drug effect for M+ trials, the number of learned words during encoding ($r_s$ = −0.45, p=0.025), the total number of remembered words during Day 2 ($r_s$ = −0.67, p<0.001) and, strikingly, the recognition rate ($r_s$ = −0.49, p=0.017), showed a significant correlation with the PAS (all correlations were FDR-corrected at a p<0.05 threshold, see *Figure 2A*). One participant was excluded from the correlations with the learning scores on Day 2 after being identified as a bivariate outlier (note that if included, the correlations become more significant: number of remembered words during Day 2, $r_s$ = −0.71, p<0.001; recognition rate, $r_s$ = −0.55, p=0.005). Importantly, this participant obtained the highest (more anhedonic) score on the PAS (score of 24, more than two standard deviations above the mean score of 11.62 of the group). Additional correlational analyses with the results of each intervention separately (risperidone, placebo, levodopa) instead of the drug effect only, further confirm a relationship between learning and memory scores and the PAS, with Spearman's *rhos* for this relationship being consistently lower than placebo for risperidone and higher than placebo for levodopa (see the Supplemental Correlational Analyses section of Appendix 1 and *Figure 2—figure supplement 1*). Although the correlation between the drug effect (calculated as the subtraction of the percentage of change from placebo of the levodopa intervention minus the percentage of change from placebo of the risperidone intervention) for pleasantness ratings and the PAS was not significant for

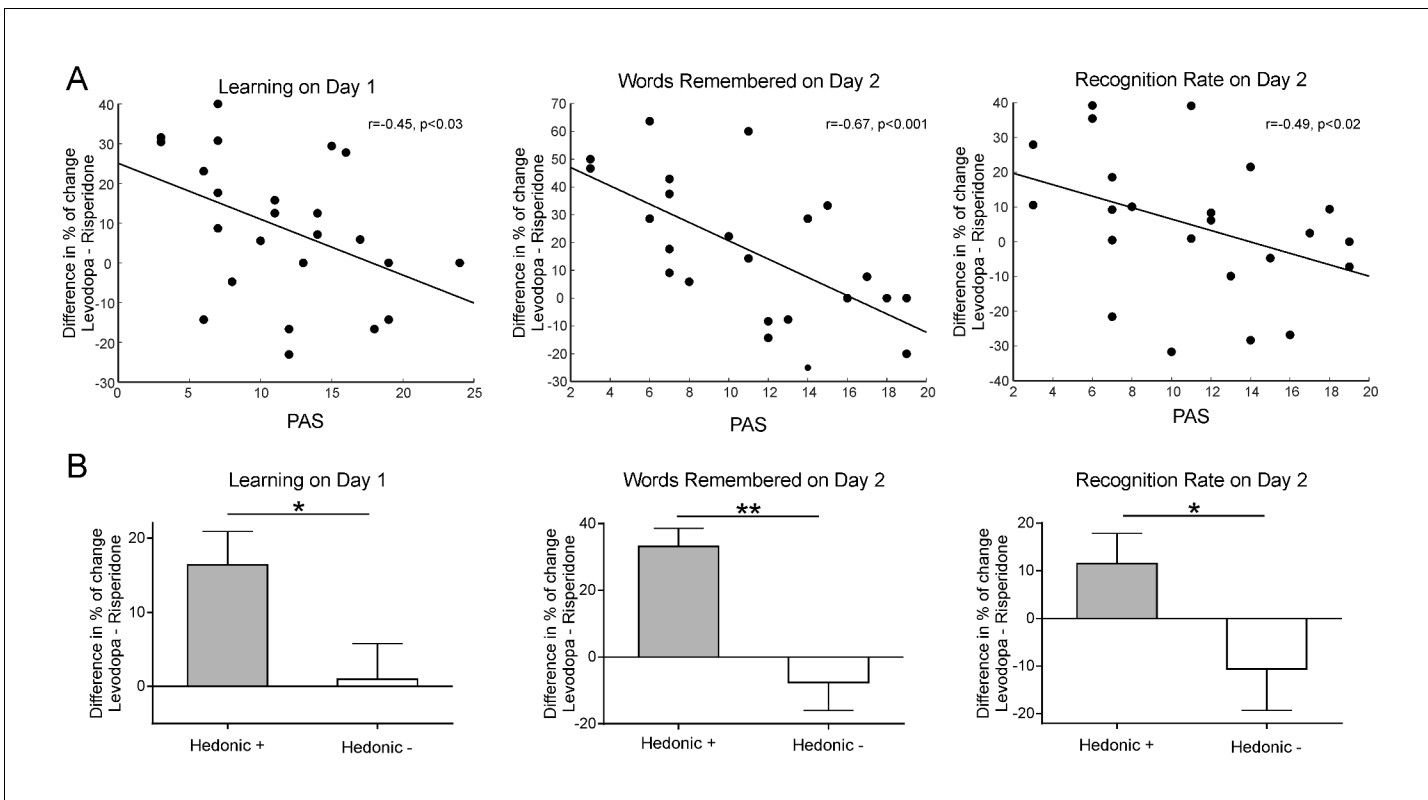

**Figure 2.** Relation between the effect of the pharmacological intervention for the M+ condition and subjective sensitivity to reward for the learning scores (i.e., online Learning on Day 1, Words Remembered on Day 2; Recognition Rate on Day 2) obtained by (**A**) correlating drug effect and PAS scores (the lower the PAS values are, the higher the general hedonia); (**B**) computing the drug effect (mean ± SEM) according to high (Hedonic +) and low (Hedonic -) hedonic subjects (median split using the PAS). *p<0.05, **p<0.001.
DOI: https://doi.org/10.7554/eLife.38113.004
The following figure supplements are available for figure 2:

**Figure supplement 1.** Correlations between the PAS and the different measures of M+ learning and memory during risperidone, placebo or levodopa interventions.
DOI: https://doi.org/10.7554/eLife.38113.005

**Figure supplement 2.** Correlations between the PAS and the M+ subjective ratings during risperidone, placebo or levodopa interventions.
DOI: https://doi.org/10.7554/eLife.38113.006

the main condition of interest (M+, $r_s$ = −0.274, p=0.196), raw pleasantness ratings during placebo and levodopa interventions separately were indeed correlated with participants' individual differences in sensitivity to reward (see the Supplemental Correlational Analyses section of Appendix 1 and *Figure 2—figure supplement 2* ).

Altogether, this pattern of results suggests that the dopaminergic pharmacological intervention induced greater memory and accuracy benefits/deficits in those participants with high sensitivity to reward. Note that the drug effect for the recognition rate, which showed no differences in memory performance when pooling all participants together, becomes significant if we divide our participants into high and low sensitivity to reward (i.e. hedonic) groups [U = 35, p=0.033, η2 = 0.19, $BF_{+0}$=2.81; groups divided according to the median split of their PAS scores; due to the reduced N of the groups we used a Mann-Whitney non-parametric statistical tests, see Materials and methods, *Figure 2B*, Appendix 1 and *Figure 1-Figure supplementary 1B*]. This difference was also present when comparing high/low groups for the number of words learned during encoding [U = 33, p=0.024, η2 = 0.21, $BF_{+0}$=4.34] and the total number of words remembered on Day 2 [U = 12.5, p<0.001, η2 = 0.49, $BF_{+0}$=78.74].

All in all, these results show that the dopaminergic pharmacological intervention did have an effect in terms of both online learning and subjective pleasantness in our learning task, inducing greater memory benefits in those participants more sensitive to reward.

## Discussion

By using a double-blind, within-subject randomized pharmacological intervention during a learning task—guided by an intrinsically regulated reward process—known to activate the SN/VTA-HP loop (*Ripollés et al., 2016*), we showed that dopamine can modulate the entrance of new information into long-term memory. In particular, the administration of a dopaminergic precursor (levodopa) and a dopaminergic antagonist (risperidone), increased and decreased, respectively, both the learning rate and the level of pleasantness experienced by the participants during encoding, as well as the number of words remembered after a consolidation period (24 hr; see *Figure 1B*). Strikingly, the memory effects induced by the dopaminergic pharmacological intervention were stronger in participants with a higher sensitivity to reward (i.e. more hedonic; see *Figure 2*).

In a previous study using the same task (*Ripollés et al., 2016*), we showed that successful learning was in itself (i.e. in the absence of external feedback) associated with increased reward processing and heightened activity within the SN/VTA and the VS. We suggested that this intrinsic reward-related signal induced a higher release of dopamine in the HP, which ultimately resulted in enhanced memory formation due to the well-known role of dopamine in mediating LTP processes. The memory effects reported here (after a 24 hr consolidation period) are also in agreement with the prediction that the SN/VTA-HP loop should specially enhance delayed memory (*Lisman and Grace, 2005*; *Lisman et al., 2011*; *Adcock et al., 2006*; *Wittmann et al., 2005*; *Wolosin et al., 2012*; *Murayama and Kitagami, 2014*). In addition to the delayed memory effects reported, here we show that dopamine had an additional role during online learning on Day 1: participants not only learned more words (i.e. they performed better) under levodopa than under risperidone, but also found the learning experience more rewarding when potentiating, rather than hindering, dopaminergic synaptic signaling. This result is in accord with previous work demonstrating that dopamine improves feedback-based learning in humans (*de Vries et al., 2010*) and also with research showing that internally generated signals of self-performance—driven by mesolimbic areas and in absence of external feedback—can guide and improve perceptual learning in humans (*Daniel and Pollmann, 2012*; *Daniel and Pollmann, 2014*; *Guggenmos et al., 2016*) and song learning (i.e. motor performance) in songbirds (*Mandelblat-Cerf et al., 2014*). Moreover, these results also converge with proposals highlighting the role of internally triggered and regulated reward signals in sustaining intrinsic motivation to perform activities that allow acquiring and storing new information (*Gottlieb et al., 2013*; *Berlyne, 1960*). Importantly, the ability to glue information acquisition activities to internal reward mechanisms might have been crucial to the evolutionary success of humans (e.g. increasing environmental control, acquiring language; *Barto, 2013*; *Ripollés et al., 2014*).

An interesting interpretation of our results is therefore that the level of dopamine directly affected the reward value or the salience (*Knecht et al., 2004*) of the learning outcome in our task (i.e. learning was more enjoyable), prompting participants to be more motivated (*Murty and*

*Adcock, 2014*) and to perform better. The VS, through its connections to the prefrontal cortex (PFC; *Lehéricy et al., 2004*; *Cummings, 1993*; *Alexander et al., 1986*), is located in a perfect anatomical position to add information about the relevance, salience and motivational value (*Berridge and Kringelbach, 2008*) of the stimuli to be learned into the SN/VTA-HP loop. As both the VS and the PFC are known to contain and receive dopaminergic receptors and projections (*Haber and Knutson, 2010*), dopamine might be able to alter this input, thus modulating the perceived reward underpinning the learning processes. This, together with the significant correlation between the PAS and the memory and learning scores (see *Figure 2*), is also in agreement with previous studies showing that anhedonia is associated with both reduced activity and connectivity between regions within the mesolimbic reward pathway (especially between the VS and the SN/VTA; *Keller et al., 2013*). An alternative explanation, which cannot be fully ruled out, is that the benefit in performance was driven by the suggested role of dopamine in working memory and attention (*Surmeier, 2007*; *Brozoski et al., 1979*; *Linssen et al., 2014*; *Drijgers et al., 2012*; *Mehta and Riedel, 2006*). However, the fact that no significant learning or memory benefits were induced in the control M- condition and, especially, the relationship between the learning improvements during the encoding phase and the participants' sensitivity to reward (for a similar effect, see *Ferreri and Rodriguez-Fornells, 2017*), suggest that the learning enhancement was partially driven by reward-related and dopamine-dependent processes (*Diehl and Gershon, 1992*; *Nieoullon and Coquerel, 2003*; *de Vries et al., 2010*).

The dopaminergic-dependent memory effects reported in this work are also in line with previous studies using risperidone and levodopa, although note that there is a lack of data on the memory effects of risperidone in healthy humans. The cognitive effects caused by the blockade of D2 receptors have usually been studied in the treatment of schizophrenia (*Rabella et al., 2016*; *Désaméricq et al., 2014*), with antipsychotic drugs (including risperidone) sometimes improving cognitive function (*Keefe et al., 1999*), but also leading to impairments in several cognitive domains (*Sakurai et al., 2013*), including executive functions and memory (*Uchida et al., 2009*; *Hori et al., 2006*). On the other hand, levodopa intake has been related to improvements in feedback-based grammar learning (*de Vries et al., 2010*), semantic activation and priming (*Angwin et al., 2004* and *Angwin et al., 2009*; *Copland et al., 2009*) and, most importantly, memory and learning (*Shellshear et al., 2015*; *Knecht et al., 2004*; *Chowdhury et al., 2012*). In this vein, the studies in which levodopa intake was related to long-term memory benefits, used associative learning tasks and suggested that the memory enhancements occurred possibly due to the increase of the levels of dopamine in the HP (*Shellshear et al., 2015*; *Knecht et al., 2004*; *Chowdhury et al., 2012*). However, our findings draw a more complex and perhaps more informative picture: the lack of a clear and significant memory enhancement for the control M- condition and the fact that more hedonic participants benefitted the most from the dopaminergic intervention only in the learning condition related to reward (M+), suggest that when using a reward-based learning task (*Apitz and Bunzeck, 2013*; *Patil et al., 2017*; *Oyarzún et al., 2016*; *Kizilirmak et al., 2016*; *de Vries et al., 2010*), the level of memory enhancement depends on dopamine synaptic signaling, but also on individual differences in sensitivity to reward (*Ferreri and Rodriguez-Fornells, 2017*; *Mas-Herrero et al., 2014*; *Camara et al., 2010*; *Marco-Pallarés et al., 2009*; *Padrão et al., 2013*). This discovery can be crucial for dopamine-related pharmacological interventions in, for example, clinical populations with language deficits (*Berthier et al., 2011*). Indeed, studies with levodopa in aphasia recovery, have resulted in both positive (*Seniów et al., 2009*) and negative (*Breitenstein et al., 2015*; *Leemann et al., 2011*) effects. In this type of therapy, in which patients try to learn of re-learn words that are no longer accessible (*Brady et al., 2012*), the intensity of the language training is usually related with recovery (*Bhogal et al., 2003*) and it has been suggested that high-training intensity may cause a ceiling effect that prevents levodopa from providing additional memory benefits (*Breitenstein et al., 2015*; *Leemann et al., 2011*). A reward-based learning task such as the one used here, along with a better understanding of the interaction between the dopaminergic precursor and the patient's hedonic state could aid to achieve a more personalized and efficient rehabilitation success, without the need for high intensive training.

In conclusion, here we show that a dopaminergic pharmacological intervention is able to modulate behavioral measures of pleasantness, task-performance and long-term memory according to inter-individual differences in reward sensitivity. These findings further advance the idea that learning—even when achieved using a task guided by intrinsic reward—is a dopamine-dependent

process, and shed new light on possible reward-based interventions for learning stimulation and/or rehabilitation.

## Materials and methods

### Participants

Around 150 individuals responded to advertisements and were contacted for a first phone pre-screening. Of those, 45 confirmed their availability and, after giving informed consent, were admitted at the hospital for further screening, medical examination and laboratory exams (blood and urine analysis). The study was approved by the Ethics Committee of Hospital de la Santa Creu i Sant Pau and the Spanish Medicines and Medical Devices Agency (EudraCT 2016-000801-35). The study was carried out in accordance with the Declaration of Helsinki and the ICH Good Clinical Practice Guidelines. All volunteers gave their written informed consent to participation prior to any procedure.

Subjects were judged healthy at screening 3 weeks before the first dose based on medical history, physical examination, vital signs, electrocardiogram, laboratory assessments, negative urine drug screens, and negative hepatitis B and C, and HIV serologies. The volunteers were excluded if they had used any prescription or over-the-counter medications in the 14 days before screening, if they had a medical history of alcohol and/or drug abuse, a consumption of more than 24 or 40 grams of alcohol per day for female and male, respectively, or if they smoked more than 10 cigarettes/day. Women with a positive pregnancy test or not using efficient contraception methods and subjects with musical training or those unable to understand the nature and consequences of the trial or the testing procedures involved were also excluded. Additionally, volunteers were requested to abstain from alcohol, tobacco and caffeinated drinks at least during the 24 hr prior to each experimental period.

Twenty nine volunteers were randomized and completed the study (19 females, mean age = 22.83 ± 4.39) in exchange of a monetary compensation according to the Spanish Legislation. The original sample size was chosen to be 30 participants, but one participant dropped out early in the study and only 29 finalized it. This sample size was selected based on several criteria, including the recommendation that, in order to achieve 80% of power, at least 30 participants should be included in an experiment in which the expected effect size is medium to large (*Cohen, 1988*). In addition, we took into account the sample sizes of previous studies using levodopa to modulate memory (range: between 10 and 30 participants; *Apitz and Bunzeck, 2013*; *Copland et al., 2009*, *de Vries et al., 2010 Knecht et al., 2004*; *Chowdhury et al., 2012*; *Shellshear et al., 2015*) and our previous behavioral studies using the same learning task (24 participants; *Ripollés et al., 2016*). We also computed a sample size analysis using the G*Power program, which showed that a sample size of 28 was required to ensure 80% of power to detect a significant effect (0.25) in a repeated-measures ANOVA with three sessions at the 5% significance level. We excluded three participants from the analyses after showing very poor memory performance on the word learning task during the placebo session (on Day 2, they remembered less than four of the M+ words learned during the encoding session). The final sample analyzed for this learning paradigm consisted of 26 participants (17 women, mean age = 22.27 ± 3.69).

### Study design and procedure

This double-blind, crossover, treatment sequence-randomized study was performed at the Neuro-psychopharmacology Unit and Center for Drug Research (CIM) of the Santa Creu i Sant Pau Hospital of Barcelona (Spain). Experimental testing took place over three sessions. For each session, participants arrived at the hospital under fasting conditions and were given a light breakfast. Subsequently, they received in a double-blind masked fashion a capsule containing the treatment (see Appendix 2 for details about counter-balancing across drug and placebo sessions): a dopaminergic precursor with an inhibitor of peripheral dopamine metabolism (levodopa, 100 mg + carbidopa, 25 mg), a dopamine receptor antagonist (risperidone, 2 mg), or placebo (lactose). The dopaminergic system has a physiological or intrinsic state whose effects are most likely reflected by the values of the dependent variables measured during the placebo session. In this study, we intended to lower and raise this baseline dopaminergic state by means of two independent pharmacological interventions

involving low-to-moderate doses of levodopa and risperidone. Drug doses were carefully chosen to be low enough to induce the desired modulation but not too large to allow collateral effects to become a confounding factor. In particular, the levodopa dose was kept in line with previous studies in healthy participants and within the dose range administered in clinical practice for the treatment of Parkinson's disease. While a higher dose could have been administered, increasing the likelihood of the dependent variables showing statistically significant differences when compared with placebo, this would have led to an unacceptable higher risk of adverse events (e.g. dangerous decreases in blood pressure, intense nausea and vomiting, prominent general discomfort). In addition, increasing the levodopa dose could also induce negative effects on cognition due to the inverted U-shaped effects of dopamine (*Chowdhury et al., 2012*). Regarding risperidone, a higher dose could have also confounded the experiment due to the well-known sedative effects risperidone can induce in healthy volunteers when administered at higher doses. Thus, drug doses use were decided upon these ethical concerns and the binding request on the part of our local Institute Review Board.

After 1 hr of completing several behavioral tasks not described in the current manuscript, the participants completed our word learning task which lasted 45 min approximately. Next, participants spent their time in a resting room and were allowed to leave the hospital after 6 hr from the treatment administration. For each session, each participant came back 24 hr after for a behavioral retesting (without any pharmacological intervention), which lasted about 15 min. At least 1 week passed between one session and the other.

## Experimental word learning task

The task was virtually identical to that of our previous work (*Ripollés et al., 2014*; *Ripollés et al., 2016*; *Ripollés et al., 2017*). Stimuli were presented using the Psychophysics Toolbox 3.09 (*Brainard, 1997*) and Matlab version R2012b. Stimuli consisted of 168 pairs of eight word-long Spanish sentences ending in a new-word, built with an increasing degree of contextual constraint (*Mestres-MisseMissé et al., 2007*; *Mestres-Missé et al., 2014*). Mean cloze probability (the proportion of people who complete a particular sentence fragment with a particular word) was $29.16 \pm 18.95\%$ for the first sentence (low constraint), and $81.67 \pm 11.80\%$ for the second (high constraint). The new-words respected the phonotactic rules of Spanish, were built by changing one or two letters of an existing word (mean number of letters = $6.02 \pm 0.99$) and always stood for a noun (mean frequency $43.26 \pm 78.94$ per million).

For each of the three different sessions, only half of the pairs of sentences disambiguated multiple meanings, thus enabling the extraction of a correct meaning for the new-word (M+ condition; e. g., 1. 'Every Sunday the grandmother went to the *jedin*'2. 'The man was buried in the *jedin*'; *jedin* means graveyard and is congruent with both the first and second sentences). For the other pairs, second sentences were scrambled so that they no longer matched their original first sentence. In this case, the new-word was not associated with a congruent meaning across the sentences (M- condition; e.g., 1. 'Every night the astronomer watched the *heutil*'. Moon is one possible meaning of *heutil*. 2. 'In the morning break co-workers drink *heutil*.' Coffee is now one of the possible meanings of *heutil*, which is not congruent with the first sentence). These constituted the M- condition in which congruent meaning extraction was not possible. To ensure that both stimulus types were equally comparable, participants were told that it was just as crucial to learn the words of the M+ condition as it was to correctly reject the new-words from the M- condition.

Given that the pharmacological intervention included three sessions, we created three versions of our task that only differed in the stimuli being presented. Thus, the 168 pairs of sentences were divided into six lists of 28 pairs (as aforementioned, two conditions, M+ and M-, were presented in each of the three sessions). The six lists were created so that there were no differences (one-way ANOVA) in the cloze probability of the sentences [first sentences: $F(5,162)=0.688$, p=0.633, $\eta2 = 0.021$, $BF_{10} = 0.044$; second sentences: $F(5,162)=0.419$, p=0.835, $\eta2 = 0.013$, $BF_{10} = 0.03$], the frequency of the meanings of the real words to be learned [$F(5,162)=1.324$, p=0.256, $\eta2 = 0.039$, $BF_{10} = 0.13$] or the total number of letters of the new-words [$F(5,162)=1.10$, p=0.360, $\eta2 = 0.033$, $BF_{10} = 0.09$]. The six lists of sentences were randomly assigned in pairs to the three different sessions. Presentation of the lists was counterbalanced across the experiments so that half of the times one list was used for the M+ condition and the other half for the M-. For each participant, new-words were randomly assigned to each pair of sentences.

During each session, four pairs of M+ and four pairs of M- sentences were presented per learning block (7 blocks in total). Therefore, a total of 28 new-words from the M+ and 28 from the M- conditions were presented during each of the three sessions. In order to achieve an ecologically valid paradigm, presentation of the first and second sentences with the same new-word at the end were separated in time. The four first sentences of each of the M+ and M- conditions (a total of eight new-words) were presented in a pseudo-randomized order (e.g., M + 1A, M-1A, M-1B, M-1C, M + 1B, M + 1C, M + 1D, M-1D). Then, the second 'pair' sentences+ and M- conditions were presented (i.e. second presentation of the identical eight new-words), again in a pseudo-randomized order (e.g. M-2C, M-2B, M + 2B, M + 2D, M-2D, M + 2C, M + 2A, M-2A). The temporal order of the different new-words during first sentence presentation was not related in any systematic way to the order of presentation of the same new-words for their second sentence. Participants were instructed to produce a verbal answer 8 s after the new-word of a second sentence appeared. If participants thought that the new-word had a congruent meaning, they had to provide its meaning in Spanish (e.g. *graveyard*). If the new-word had no consistent meaning, they had to say the word *incongruent.* If they did not know whether the new-word had a consistent meaning or not, they had to remain silent. Vocal answers were recorded and later corrected (for the M+ condition, incorrect answers included misses, providing the wrong meaning or saying *incongruent;* for the M- condition, incorrect answers included misses or providing any meaning at all). After giving a verbal answer, participants first provided a confidence rating that allowed for the assessment of the subjective evaluation of their performance. Specifically, subjects were requested to enter, using the keyboard, a value between −4 and 4 (9 point scale with 0 as the neutral value). Then, participants had to rate their emotions with respect to arousal and pleasantness using the 9-point (as with confidence ratings, from −4 to 4) visual *Self-Assessment Manikin* scale (SAM). For valence/pleasantness, the SAM ranges from a sad, frowning figure (i.e. very negative) to a happy, smiling figure (i.e. very positive). For arousal, the SAM ranges from a relaxed figure (i.e. very calm) to an excited figure (i.e. very aroused). All participants completed a training block to familiarize them with the task.

Each trial started with a fixation cross lasting 1000 ms, continued with the seven first Spanish words of the sentence presented for 2 s, and was followed by a 1-s duration dark screen. The new-word was presented for 1000 ms and was followed by 7 s of a small fixation point presented in the middle of the screen. For first sentences, a new trial was presented after 3 s of dark screen. For second sentences, after this period, a screen with the word *Answer* appeared and subjects had 3 s to produce a verbal answer. Then, the confidence and SAM scales for pleasantness and arousal were sequentially presented (the experiment did not continue until participants provided a rating). Finally, a new second sentence trial started after 3 s of dark screen. All words were placed in the middle of a black screen with a font size of 22 and in white color.

To avoid biasing our results, participants were not told at any point prior to the start of the experiment that the goal of the study was to assess whether the learning of a new-word and its meaning was intrinsically rewarding. Instead, they were told that the objective of the study was to assess how reading load affects mood and that, in order to ensure that there was a real reading load, they had to learn the words of the M+ condition and to detect the incongruence of the new-words from the M-. Finally, participants were told that they had to give pleasantness and arousal ratings when the second sentences appeared because that moment signaled that reading load had already occurred (i.e. half of the encoding block had already elapsed). After the experiment, participants were first questioned about the objective of the study. None of them answered that it was to assess whether word-learning was rewarding.

Approximately 24 hr after the learning lesson ended, participants returned to the lab to complete a recognition test (note that no drugs were administered to subjects on Day 2). In this test, participants were presented, in a pseudo-randomized order, with all the 28 M+ and 28 M- new-words used during the encoding session. This test was devised in order to assess which of the learned words during encoding were still remembered and which of them had been forgotten after a 24 hr retention period. Participants were aware that they would complete this test before completing the encoding session. It was made explicit that they would assess both M+ and M- new-words during the test phase. In the test, participants were presented with a new-word at the centre of the screen with two possible meanings below: one on the left and one on the right. If the new-word tested did not have a congruent meaning associated between the first and the second sentence, and thus correct meaning extraction was not possible (M- condition), participants had to press a button located

in their left hand. In this case, the two possible meanings presented served as foils: one was the meaning evoked by the second sentence of the M- new-word being tested; the other word shown was the meaning evoked by another second sentence presented in the same run as the new-word being tested. Instead, if the new-word tested had a consistent meaning through the first and second sentence, and thus correct meaning extraction was possible (M+ condition), participants had to select the correct meaning among the two presented. In this case, one of the two possible meanings was correct and the other, which served as a foil, was the meaning of another new-word presented in the same run. In addition, participants could also press a fourth button if they did not know the answer. Thus, chance level was at 25% (no consistent meaning, consistent meaning on the left, consistent meaning on the right, not remembered).

## Statistical analyses for confidence, pleasantness and arousal subjective scales and learning scores for encoding and retrieval

We first assessed whether the results of the placebo session replicated our previous behavioral data (Experiment 3 in *Ripollés et al., 2016*). Besides the three subjective ratings, for these first comparisons, we used two additional scores: the percentage of words learned on Day 1 (total number of words learned divided by the total number of words presented) and the recognition rate (total number of words learned during encoding and remembered on Day 2 divided by the number of words learned during Day 1). For the analyses regarding the subjective scales, we divided our M+ trials into those in which subjects learned the new-word during the learning session and still remembered it in the test after the recognition test (*remembered* condition) and those in which the new-word was not correctly identified in the 24 hr test (*forgotten* condition). We used the same approach to divide the M- trials into those in which a word was correctly marked as incongruent during encoding and still correctly rejected after 24 hr and those in which the new-word was not correctly rejected in the follow-up test. To replicate our previous results, we first used paired t-tests to compare whether ratings for confidence, arousal and pleasantness were greater for remembered than for forgotten M + and M- new-words. We then submitted both the ratings and the learning scores to a mixed repeated measures ANOVA with Condition (M+,M-) as a within-subjects variable and Group (Pharmacological Group, Exp. 3 in *Ripollés et al., 2016*) as a between subjects variable.

Given that current behavioral results replicate our previous work (see results) and that in our previous study (*Ripollés et al., 2016*) remembered M+ words were the trials showing the highest fMRI activity within the SN/VTA-HP loop, the largest physiological response and the highest subjective pleasantness ratings, we focused all the analyses regarding the effect of the pharmacological intervention in the trials in which a word was learned during encoding and still remembered during the recognition test at 24 hr (M+ condition). For the control condition, we used those M- trials in which a word was correctly rejected during both encoding and the follow-up test. As measures for memory effects, we used the total number of words learned during encoding and remembered on the follow-up test and the percentage remembered words in the recognition test compared to the number of learned words during the learning phase (i.e. the recognition rate). Note that these two measures are *delayed memory* scores, while the percentage of learned words during Day 1 is an online learning rather than a pure delayed memory measure (participants try to learn the meaning of each new word from the two sentences presented and then provide a verbal answer when the second sentence is presented).

To ensure that no online learning or memory effects were present across sessions independent of the treatment (i.e. participants could learn more on the third session just because they completed the task twice before), we performed a control analysis for the percentage of learned words during Day 1, total number of words remembered on Day 2 and the recognition test. For each score, we computed a repeated measures ANOVA with Session (First, Second, Third; regardless of the treatment) as the main factor. There was not a significant effect of session order regardless of the treatment for the main M+ learning condition [percentage of learned words during Day 1: $F_{(2,50)}=0.957$, $p=0.391$, $\eta2 = 0.037$, $BF_{10} = 0.236$; total number of words remembered on Day 2: $F_{(2,50)}=0.489$, $p=0.616$, $\eta2 = 0.019$, $BF_{10} = 0.161$; recognition rate on Day 2: $F_{(2,50)}=0.263$, $p=0.770$, $\eta2 = 0.010$, $BF_{10} = 0.133$] or M- control condition [percentage of correctly rejected words during Day 1: $F_{(2,50)}=0.964$, $p=0.388$, $\eta2 = 0.037$, $BF_{10} = 0.234$; total number of correctly rejected words on Day 2: $F_{(2,50)}=1.250$, $p=0.295$ $\eta2 = 0.048$, $BF_{10} = 0.284$; recognition rate on Day 2: $F_{(2,50)}=0.604$, $p=0.550$, $\eta2 = 0.024$, $BF_{10} = 0.176$].

The purpose of this study was to elucidate whether *modulation of the dopaminergic system* influenced the variables under study (i.e. memory, learning and reward), rather than to assess the capacity of the drugs themselves to block or enhance the natural physiological responses influenced by dopamine. Levodopa and risperidone were chosen to 'displace' the baseline physiological system in opposite directions: risperidone to lower the effects of physiological dopamine release and levodopa to enhance dopaminergic neurotransmission. Thus, as the objective was to bring the dopaminergic system away from its intrinsic state (i.e. the placebo session) and in opposite directions, our analyses focused in directly comparing the risperidone and levodopa data against each other by using the placebo session as a baseline (but see Appendix 1 for analyses taking into account the three pharmacological sessions separately). For each learning score and subjective scale, we calculated the percentage of change from the placebo session [e.g. (levodopa score - placebo score)/(placebo score)]. Therefore, for each participant, learning score and subjective scale, we obtained the percentage of change from placebo of the risperidone and levodopa interventions. We used paired t-tests to calculate whether the difference between the changes induced by the risperidone and levodopa interventions were significant.

For the correlations between the learning scores and the PAS, we used Spearman's rho with a $p < 0.05$ FDR correction to account for the three different correlations calculated per condition. The PAS is one of the most easy to administer and well-validated tests for measuring reward sensitivity and anhedonia. In addition, it shows convergent results with more modern tests measuring trait anhedonia, as the Mood and Anxiety Symptom Questionnaire Short Form (*Keller et al., 2013*) or the Temporal Experience of Pleasure Scale (*Favrod et al., 2009*). Some other psychometric measures of reward sensitivity and anhedonia that are translated and validated into Spanish, such as the Sensitivity to Punishment/Sensitivity to Reward Questionnaire (*Torrubia et al., 2001*) or the BIS/BAS (*Caseras et al., 2003*) show poorer alpha coefficients—less than 0.78 and 0.73, respectively—than the PAS (0.92; *Fonseca-Pedrero et al., 2009*). Indeed, we have previously used the PAS in previous studies exploring individual differences in anhedonia trait (*Padrão et al., 2013*; *Mas-Herrero et al., 2014*; *Martínez-Molina et al., 2016*; *Mas-Herrero et al., 2018*). Here, the PAS was used as a proxy to reflect the degree of pleasure taken by individuals when engaging in rewarding behavior (*Der-Avakian and Markou, 2012*). Note that two participants were excluded from this analysis as they did not complete the PAS. We also correlated the learning scores with a weight-dependent measure of drug dose, calculated in mg of levodopa/risperidone per kilogram. Finally, we used the median PAS value to split our final sample of 24 participants into high and low hedonic groups. For the learning scores, we first calculated the drug effect as a subtraction of the percentage of change from placebo induced by the levodopa intervention minus that induced by the risperidone intervention. We then assessed were the total drug effect for the learning scores was different for high vs. low hedonic groups by using a non-parametric test Mann-Whitney (to better account for the reduced number of participants in each group).

For significant interactions of mixed between-within ANOVA models, partial eta squares (η2) is provided as a measure of effect size. For significant differences in between group one-way ANOVAs, eta squares (η2) is provided (calculated by dividing the between groups sum of squares by the total sum of squares). For significant differences measured with t-tests, Cohen's d is provided after applying Hedges' correction (the average of the standard deviation of the variables being compared was used as a standardizer; Cumming, 2012). For significant differences measured with the Mann-Whitney test, eta squares (η2) is provided (calculated as $Z^2/N$)

In addition, confirmatory Bayesian statistical analyses were computed with the software JASP using default priors (*JASP Team, 2018*; *Morey et al., 2015*; *Rouder and Morey, 2012*; *Wagenmakers et al., 2018b*; *Wagenmakers et al., 2018a*). We reported Bayes factors ($BF_{10}$), which reflect how likely data is to arise from one model, compared, in our case, to the null model (i.e. the probability of the data given H1 relative to H0). For comparisons with a strong a priori, the alternative hypothesis was specified so that one group/condition was greater than the other ($BF_{+0}$). We did this, specifically, for the drug effects comparisons in which we expected levodopa and risperidone to facilitate and disrupt learning/ratings, respectively; and for the group comparisons in which we expected more hedonic participants to remember more words than less hedonic participants. For mixed within-between models we used the Bayes Inclusion factor based on matched models, representing the evidence for all models containing a particular effect to equivalent models stripped of that effect ($BF_{Inlcusion}$, also called Baws factor).

## Acknowledgements

We thank the staff of the Centre d'Investigació del Medicament de l'Institut de Recerca HSCSP for their help. The present project has been funded by the Spanish Government (MINECO Grant PSI2011-29219 to ARF and AP2010-4179 to PR) and German Research Council (DFG-SFB-779/A15 to TN). LF was supported by Morelli-Rotary postdoctoral fellowship. MV was partially supported by FIS trough grant CP04/00 121 from the Spanish Health Ministry in collaboration with Institut de Recerca de l'Hospital de la Santa Creu i Sant Pau, Barcelona; she is a member of CIBERSAM (funded by the Spanish Health Ministry, Instituto de Salud Carlos III). This study has been funded by Ministerio de Economía, Industria y Competitividad (MINECO), which is part of Agencia Estatal de Investigación (AEI), through the project PSI2015-69178-P (co-funded by European Regional Development Fund. ERDF, a way to build Europe).

## Additional information

### Funding

| Funder | Grant reference number | Author |
| --- | --- | --- |
| Ministerio de Economía y Competitividad | AP2010-4179 | Pablo Ripollés |
| Morelly-Rotary Postdoctoral Fellowship | | Laura Ferreri |
| Ministerio de Sanidad, Servicios Sociales e Igualdad | CP04/00 121 | Marta Valle |
| Deutsche Forschungsgemeinschaft | DFG-SFB-779/A15 | Toemme Noesselt |

The funders had no role in study design, data collection and interpretation, or the decision to submit the work for publication.

### Author contributions

Pablo Ripollés, Conceptualization, Software, Formal analysis, Investigation, Methodology, Writing—original draft, Writing—review and editing; Laura Ferreri, Conceptualization, Data curation, Formal analysis, Investigation, Methodology, Writing—original draft, Project administration; Ernest Mas-Herrero, Data curation, Formal analysis, Methodology, Writing—review and editing; Helena Alicart, Data curation, Investigation, Writing—review and editing; Alba Gómez-Andrés, Data curation, Writing—review and editing; Josep Marco-Pallares, Formal analysis, Writing—review and editing; Rosa Maria Antonijoan, Data curation, Methodology, Writing—review and editing; Toemme Noesselt, Formal analysis, Methodology, Writing—review and editing; Marta Valle, Data curation, Investigation, Methodology, Project administration, Writing—review and editing; Jordi Riba, Conceptualization, Supervision, Investigation, Methodology, Project administration, Writing—review and editing; Antoni Rodriguez-Fornells, Conceptualization, Supervision, Funding acquisition, Methodology, Project administration, Writing—review and editing

### Author ORCIDs

Pablo Ripollés (ID) http://orcid.org/0000-0002-8463-3723
Toemme Noesselt (ID) https://orcid.org/0000-0002-9611-9713

### Ethics

Human subjects: This study was performed according to local ethics and to the Declaration of Helsinki. It was approved by the Ethics Committee of Hospital Sant Pau and by the Spanish Medicines and Medical Devices Agency (EudraCT 2016-000801-35). All participants gave informed written consent and received compensation for their participation in the study according to Spanish legislation.

### Decision letter and Author response

Decision letter https://doi.org/10.7554/eLife.38113.016

Author response https://doi.org/10.7554/eLife.38113.017

## Additional files

### Supplementary files

• Transparent reporting form
DOI: https://doi.org/10.7554/eLife.38113.007

### Data availability

Data is available via Dryad (https://dx.doi.org/10.5061/dryad.g5f7v1j)

The following dataset was generated:

| Author(s) | Year | Dataset title | Dataset URL | Database, license, and accessibility information |
|---|---|---|---|---|
| Ripollés P, Ferreri L, Mas-Herrero E, Alicart H, Gómez-Andrés A, Marco-Pallares J, Antonijoan R, Noesselt T, Valle M, Riba J, Rodriguez-Fornells A | 2018 | Data from: Intrinsically regulated learning is modulated by synaptic dopamine signaling | https://dx.doi.org/10.5061/dryad.g5f7v1j | Available at Dryad Digital Repository under a CC0 Public Domain Dedication |

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

# Appendix 1

DOI: https://doi.org/10.7554/eLife.38113.008

## Supplemental behavioural analyses

While the purpose of this work was to elucidate whether modulation of the dopaminergic system influenced the variables under study (i.e., intrinsically regulated learning), rather than to assess the capacity of the drugs themselves to significantly block or enhance the natural physiological responses influenced by dopamine, here we provide separated analyses for the three different sessions (risperidone, placebo, levodopa) and the main condition of interest (M +). Specifically, we used a repeated measures ANOVA with Intervention as a within subject factor (Risperidone, Placebo and Levodopa) and paired post-hoc t-tests to re-analyze the data for the learning and memory scores and the subjective ratings. For the learning measures, taking into account the relationship between the PAS and the percentage of learned and remembered words (see *Figure 2* in the main text), we also calculated a 3 × 2 mixed repeated measures ANOVAs with Intervention (Risperidone, Placebo and Levodopa) as the within and Group (H+, H-) as the between group factor.

For the main M + learning condition, we found a main effect of Intervention for the percentage of words learned on Day 1 [$F(2,50)=5.041$, $p=0.01$, partial $\eta 2 = 0.168$, $BF_{10} = 4.524$] and the total number of remembered words on Day 2 [$F(2,50)=4.793$, $p=0.012$, partial $\eta 2 = 0.161$, $BF_{10} = 3.667$; see *Figure 1—figure supplement 1A*]. Post-hoc paired t-tests with Holm-Bonferroni correction showed, as expected, a significant difference between the levodopa and the risperidone interventions for both online learning on Day 1 [$t(25)=2.95$, $p\_corr = 0.02$, $d = 0.580$, $BF_{10} = 6.636$] and the total number of words remembered on Day 2 [$t(25)=2.44$, $p\_corr = 0.044$, $d = 0.479$, $BF_{10} = 2.452$]. Comparisons of levodopa against placebo were not significant after the correction for multiple comparisons was applied [online Learning on Day 1: $t(25)=1.629$, $p\_corr = 0.214$, $d = 0.319$, $BF_{10} = 0.663$; Total Number of Remembered Words on Day 2: $t(25)=-0.240$, $p\_corr = 0.812$, $d = 0.047$, $BF_{10} = 0.213$]. The risperidone vs. placebo comparison reached significance for the total number of remembered words on Day 2 [$t(25)=-2.90$, $p\_corr = 0.023$, $d = 0.569$, $BF_{10} = 5.943$], while was still not significant for the percentage of learned words on Day 1 [$t(25)=-1.63$, $p\_corr = 0.214$, $d = 0.328$, $BF_{10} = 0.704$]. As expected, the recognition rate on Day 2 yielded no significant main effect of Intervention [$F(2,50)=1.661$, $p=0.20$, partial $\eta 2 = 0.062$, $BF_{10} = 0.395$].

When dividing the participants into two groups according to their PAS scores (H+, H-), a significant Group x Intervention interaction was found for the total number of remembered words on Day 2 [$F(2,50)=7.825$, $p<0.001$, partial $\eta 2 = 0.262$, $BF_{Inclusion} = 25.747$], while for the percentage of words learned on Day 1 [$F(2,50)=2.786$, $p=0.073$, partial $\eta 2 = 0.112$, $BF_{Inclusion} = 1.204$] and the recognition rate on Day 2 [$F(2,50)=2.615$, $p=0.084$, partial $\eta 2 = 0.106$, $BF_{Inclusion} = 1.047$] the interaction was marginally significant (see *Figure 1—figure supplement 1B*).

Regarding the subjective ratings, as expected, there was a significant main effect of Intervention for Pleasantness [$F(2,50)=5.25$, $p=0.009$, partial $\eta 2 = 0.174$, $BF_{10} = 5.15$] and Confidence [$F(2,50)=3.70$, $p=0.032$, partial $\eta 2 = 0.129$, $BF_{10} = 1.76$], but not for Arousal [$F(2,50)=0.319$, $p=0.728$, partial $\eta 2 = 0.013$, $BF_{10} = 0.141$; see *Figure 1—figure supplement 1C*]. Post-hoc paired t-tests with Holm-Bonferroni correction showed that Pleasantness ratings were higher under levodopa than under risperidone [$t(25)=2.85$, $p\_corr = 0.026$, $d = 0.560$, $BF_{10} = 5.378$], with no significant differences between the levodopa [$t(25)=1.584$, $p\_corr = 0.137$, $d = 0.311$, $BF_{10} = 0.623$] or risperidone interventions [$t(25)=-1.903$, $p\_corr = 0.137$, $d = 0.373$, $BF_{10} = 0.985$] against the placebo one. For Confidence ratings no significant differences were found between any pair of interventions after applying the correction for multiple comparisons [risperidone vs. placebo: $t(25)=-2.275$ $p\_corr = 0.095$, $d = 0.446$, $BF_{10} = 1.816$; risperidone vs. levodopa: $t(25)=2.231$, $p\_corr = 0.095$, $d = 0.438$, $BF_{10} = 1.683$; levodopa vs. placebo: $t(25)=0.521$, $p\_corr = 0.607$, $d = 0.102$, $BF_{10} = 0.235$].

The significant main effects and interactions for both memory and reward measures further demonstrated that: i) the pharmacological intervention attained its main objective, which was to bring the dopaminergic system away from its intrinsic state (i.e., the placebo intervention) and in opposite directions (see, for example the Pleasantness ratings in *Figure 1—figure supplement 1C*); ii) that intrinsically regulated learning is a dopamine-dependent mechanism; and iii) that subject-specific reward/hedonic sensitivity drastically alters learning success (modulations for memory and learning scores only happened in highly hedonic individuals, see *Figure 1—figure supplement 1B*).

## Supplemental correlational analyses

Following the same rationale as with the supplemental behavioral analyses above and taking into account the significant correlations found between the drug effect (calculated as the subtraction of the percentage of change from placebo of the levodopa intervention minus the percentage of change from placebo of the risperidone intervention) and the PAS (see *Figure 2* in the main text), here we also calculated correlations (using Spearman´s rho) between the PAS and all measures of learning, memory and the subjective ratings at each intervention separately (a total of 18 correlations; we used a $p < 0.05$ FDR corrected threshold to control for the multiple correlations calculated).

Significant correlations (FDR corrected for multiple comparisons) between the PAS and the percentage of learned words during Day 1, the total number of remembered words and the recognition rate on Day 2 were found for the levodopa intervention (see *Figure 2—figure supplement 1*). Regarding the subjective ratings, pleasantness was significantly correlated to the PAS during the placebo and levodopa interventions (see *Figure 2—figure supplement 2*). Confidence ratings were only significantly correlated to the PAS for the levodopa intervention, while, as expected, no significant associations were found for arousal scores.

In addition, *r* scores were consistently lower than placebo for risperidone and higher than placebo for levodopa. This implies that the relationship between the PAS and the different measures of memory and learning and the pleasantness ratings was also modulated by the drug intervention. These results further suggest that: i) the pharmacological intervention was able to 'displace' the baseline physiological system in opposite directions, making the learning experience more (levodopa) and less (risperidone) rewarding; ii) that this led to higher and lower learning and retention rates; and iii) more hedonic participants were able to better capitalize in the increase of the 'rewarding experience' (see, for example, the high correlation between the PAS and the retention rate on Day 2 in the levodopa intervention in to *Figure 2—figure supplement 2*).

## Individual variability of the drug effect

As can be seen in *Figure 2A*, there were some participants who performed worse in terms of learning and memory on levodopa than risperidone for the main learning condition M+ (i.e., showed a negative drug effect, calculated as the subtraction of the percentage of change from placebo of the levodopa session minus the percentage of change from placebo of the risperidone session). In particular, the number of participants who performed worse in the levodopa than in the risperidone session was 6 for the percentage of learned words on Day 1, 6 for the total number of words remembered on Day 2, 9 for the recognition rate on Day 2 and 9 for the pleasantness ratings. However, there was not a single participant who showed a negative drug effect in all three learning/memory scores and the pleasantness ratings at the same time. In contrast, participants showed positive values in all memory and learning scores and in the pleasantness ratings. In this vein, the drug effect for pleasantness did not significantly correlate with any of the drug effects for learning and memory scores (learning on Day 1: $r = -0.04$, $p = 0.829$; total number of remembered words on Day 2: $r = 0.108$, $p = 0.608$; recognition rate on Day 2: $r = 0.153$, $p = 0.463$).

If we assess the overlap for the learning and memory scores on their own, there was only one participant showing a negative drug effect (levodopa minus risperidone) in the percentage of learned words on Day 1, the total number of words remembered on Day 2 and the recognition rate on Day 2 at the same time. In contrast, 12 showed positive values on the

three measures. Indeed, while the drug effect for the percentage of learned words on Day 1 correlated with the drug effect for the total number of words remembered on Day 2 ($r = 0.569$, $p<0.003$), the relationship was not significant for the recognition rate on Day 2 ($r = -0.084$, $p=0.689$). Finally, 4 participants performed poorer in the levodopa than the risperidone session for both the total number of words remembered on Day 2 and the recognition rate on Day 2, while 15 participants showed positive values in both measures. Indeed, the drug effects for these two memory measures are significantly correlated ($r = 0.638$, $p<0.001$).

# Appendix 2

DOI: https://doi.org/10.7554/eLife.38113.009

## Counterbalancing across treatments

Treatment administration was randomized and balanced as shown in *Appendix 2—table 1*. Treatment-letter assignment was performed randomly by a member of the Biometrics department of Sant Pau Hospital, who kept the record unavailable to the investigators until finalization of the experimental sessions.

**Appendix 2—table 1.** Counterbalancing across treatments, with six different sequences of letters randomly assigned to N = 30. A corresponded to risperidone, B to placebo and C to levodopa.

| Volunteer | Sequence | Treatment order |
|---|---|---|
| 1 | 5 | B/A/C |
| 2 | 4 | A/C/B |
| 3 | 5 | B/A/C |
| 4 | 6 | C/B/A |
| 5 | 2 | B/C/A |
| 6 | 2 | B/C/A |
| 7 | 2 | B/C/A |
| 8 | 6 | C/B/A |
| 9 | 1 | A/B/C |
| 10 | 5 | B/A/C |
| 11 | 3 | C/A/B |
| 12 | 5 | B/A/C |
| 13 | 3 | C/A/B |
| 14 | 4 | A/C/B |
| 15 | 1 | A/B/C |
| 16 | 1 | A/B/C |
| 17 | 2 | B/C/A |
| 18 | 5 | B/A/C |
| 19 | 1 | A/B/C |
| 20 | 3 | C/A/B |
| 21 | 4 | A/C/B |
| 22 | 6 | C/B/A |
| 23 | 4 | A/C/B |
| 24 | 6 | C/B/A |
| 25 | 6 | C/B/A |
| 26 | 3 | C/A/B |
| 27 | 1 | A/B/C |
| 28 | 4 | A/C/B |
| 29 | 2 | B/C/A |
| 30 | 3 | C/A/B |

DOI: https://doi.org/10.7554/eLife.38113.010

