## [Decision Letter]

Thank you for submitting your article "Intrinsically regulated learning is modulated by synaptic dopamine availability" for consideration by *eLife*. Your article has been reviewed by three peer reviewers, including Vishnu Murty as the Reviewing Editor and Reviewer #1, and the evaluation has been overseen by Sabine Kastner as the Senior Editor. The following individual involved in review of your submission has agreed to reveal their identity: Bart Larsen (Reviewer #3).

The reviewers have discussed the reviews with one another and the Reviewing Editor has drafted this decision to help you prepare a revised submission.

Summary:

Rippolles and colleagues present an interesting pharmacological study following-up on prior work showing engagement of VTA-HPC loops on intrinsically regulated learning. The authors use a pharmacological challenge to both up-regulate (l-Dopa) and down-regulate (risperidone) dopamine signaling during learning, and characterize the effects of putative "intrinsic rewards" on encoding, delayed memory, and affective ratings. Myself and the two reviewers all found merit in the manuscript, and were excited about the contribution of this pharmacological challenge on a growing field of reward and episodic memory.

Essential revisions:

1) The Introduction/Discussion warrants more theoretical and empirical background on l-Dopa and Risperidone. In particular, the authors should address the differences across these pharmacological challenges as they relate to dopamine physiology (i.e., non-selective increases vs. D2-specific antagonism). Further, the authors should provide more details on how their findings fit in with prior work using these drug manipulations during learning and memory.

2) All three reviewers agreed that the authors should include further analysis comparing each drug group against placebo separately. This could either be done as post-hoc t-tests or including all three factors in an ANOVA. This approach would also be relevant for the correlations with PAS (both analysis and figures), as it is unclear whether differences are driven by l-DOPA increases or risperidone decreases.

3) More details are needed on the PAS scores, including empirical and theoretical evidence for using these measures in the Introduction/Discussion. Second, can the authors comment if any individuals classify as being clinically anhedonic, and if so this should be addressed in the discussion. Third, do PAS scores relate to learning/memory in the placebo condition? Fourth, can the authors provide more details of how they believe anhedonia relates to their learning mechanisms in the discussion.

4) Did some individuals perform worse on levodopa than risperidone. If so are these the same individuals across measures of learning, delayed recognition, and affective responses. Addressing this component, in light of the complexity of DA systems, may help connect these findings to the broader literature (i.e., U-shaped DA curves).

[Editors' note: further revisions were requested prior to acceptance, as described below.]

Thank you for resubmitting your work entitled "Intrinsically regulated learning is modulated by synaptic dopamine signaling" for further consideration at *eLife*. Your article has been reviewed by three peer reviewers, one of whom is a member of our Board of Reviewing Editors, and the evaluation has been overseen by Joshua Gold as the Senior Editor.

The manuscript has been significantly improved but there are a few remaining issues that need to be addressed before acceptance, as outlined below:

As you can see below, there are just a few minor, easily addressable points. These are centered on the following items. First, the authors should qualify the discussion of the results, in particular describing them as fully bidirectional, as a few of the results did not show significant differences across baseline for both drugs. Second, the authors should revise their supplemental across-subject analysis to use each drug separately compared against baseline, as opposed to each correlation on its own. Third, the authors should include the findings (which were detailed in the revision) in the supplement of the number of participants in the L-dopa condition performing worse. Finally, the authors should include additional details of why PAS/anhedonia was used as the measure of reward sensitivity as opposed to other measures.

Below you can see the point-by-point responses by each reviewer.

Reviewer #1:

The authors did a commendable job responding to the revisions in the previous manuscript. I only had a few remaining concerns that should be easily addressable.

Firstly, the authors throughout the discussion refer to the bi-directionality of the modulation of dopaminergic neurotransmission. Given that some of the findings from the comparisons against placebo were not significant, these claims should be qualified.

Further, I believe the request for separate across-subject analyses were mis-interpreted in the first round of reviews. Rather, then looking at each correlation in each group separately, I was hoping to see the correlations with scores calculated as Risperidone>Baseline and then separately L-Dopa>Baseline. I believe these will be more compelling correlations as they remove any subject-specific variance.

Reviewer #2:

1) I think the authors could have done a better job addressing rationale for using the PAS and could include more in the discussion about this. There are many measures of reward sensitivity and anhedonia and there is not much discussion of why this measure was included.

2) Please include the findings about some people performing worse in L-dopa in the supplement.

3) I apologize for not noticing this before, so this is something for the researchers to consider in the future. There are findings that suggest that individuals who smoke cigarettes who have abstained from smoking have blunted reward anticipation to money (Sweitzer et al., Biological Psychiatry, 2014). In the future I would encourage the authors to perhaps recruit only non-smokers or to include smoking status as a variable in their analyses. I would be curious to know how many participants in their pool were smokers and if their findings differed based on smoking status. I do not think it is necessary to include in the present manuscript.

Reviewer #3:

The changes made by the authors have sufficiently addressed my concerns. In particular, I feel the added analyses and figures provide a richer presentation of the data (even though many are supplemental) substantially improve the manuscript.

---

## [Author Response]

Essential revisions:1) The Introduction/Discussion warrants more theoretical and empirical background on l-Dopa and Risperidone. In particular, the authors should address the differences across these pharmacological challenges as they relate to dopamine physiology (i.e., non-selective increases vs. D2-specific antagonism). Further, the authors should provide more details on how their findings fit in with prior work using these drug manipulations during learning and memory.

With hindsight, we agree with the reviewer that more information regarding how levodopa and risperidone affect dopamine physiology should be included in the manuscript. The Introduction section has been updated accordingly:

“Levodopa is rapidly taken up by dopaminergic neurons, transformed into dopamine and stored in vesicles from which it will be released into the synaptic cleft each time the neuron fires. […] Each of the two experimental sessions involving active drugs were intended to shift dopaminergic neurotransmission away from each individual’s physiological status, as measured in a placebo session, and in opposite directions: levodopa to enhance the dopamine availability for release into the synapse, and risperidone to reduce synaptic transmission of the dopamine-associated signal by hindering dopamine-receptor interactions (see e.g. Rabella et al., 2016; Wittman and D’Esposito, 2015; De Vries et al., 2010; Knecht et al., 2004 for the use of levodopa or risperidone during cognitive tasks).”

Regarding how our findings are related to prior work using these drug manipulations, the Discussion section has been extended to include a more detailed account of previous and current work:

Discussion section:

“The dopaminergic-dependent memory effects reported in this work are in line with previous studies using risperidone and levodopa, though note that there is a lack of data on the memory effects of risperidone in healthy humans. […] In particular, the lack of a clear and significant memory enhancement for the control M- condition and the fact that more hedonic participants benefitted the most from the dopaminergic intervention only in the learning condition related to reward (M+), suggest that when using a reward-based learning task (Apitz and Bunzeck, 2013; Patil et al., 2016; Oyarzun et al., 2016; Kizilirmak et al., 2015; de Vries et al., 2010), the level of memory enhancement depends on dopamine synaptic signaling, but also on the individual differences in sensitivity to reward (Ferreri et al., 2017; Mas-Herrero et al., 2014; Camara et al., 2010; Marco-Pallares et al., 2009; Padrao et al., 2013).”

We have also updated the title of the manuscript to reflect a more accurate account of the effects induced by risperidone and levodopa:

Intrinsically regulated learning is modulated by synaptic dopamine signaling

2) All three reviewers agreed that the authors should include further analysis comparing each drug group against placebo separately. This could either be done as post-hoc t-tests or including all three factors in an ANOVA. This approach would also be relevant for the correlations with PAS (both analysis and figures), as it is unclear whether differences are driven by l-DOPA increases or risperidone decreases.

The reviewers raise a crucial point that we now highlight in the revised version of our manuscript. While previous studies have mainly focused on the role of levodopa in enhancing memory, here we decided to use a two-way approach to assess whether our intrinsic-reward learning task was modulated bidirectionally to link dopamine-related modulation even more tightly to the behavioral read-outs and the critical receptor types.

The dopaminergic system has a physiological or intrinsic state whose effects are reflected by the values of the dependent variables measured in the course of the placebo session. What was intended and attained in our study was to lower and raise this baseline dopaminergic state by means of two independent pharmacological interventions involving low-to-moderate doses of two different drugs. These drugs were chosen to "displace" the baseline physiological system in opposite directions: risperidone to lower the effects of physiological dopamine release and levodopa to enhance dopaminergic neurotransmission. Thus, by bringing the dopaminergic system away from its intrinsic state and in opposite directions, our analyses were focused in directly comparing the risperidone and levodopa data against each other. In other words, the purpose of this study was not to assess the capacity of the drugs themselves to block or enhance the natural physiological responses influenced by dopamine, but rather to elucidate whether modulation of the dopaminergic system influenced the variables under study (i.e., memory, learning and reward).

While we hypothesize that larger doses of each treatment would have increased the reported effects, the doses of both levodopa and risperidone were carefully chosen to be low enough to induce the desired modulation but not too large to allow collateral effects to become a confounding factor. In particular, the levodopa dose was kept in line with previous studies in healthy participants and within the dose range administered in clinical practice for the treatment of Parkinson's disease (PD). While a higher dose could have been administered, again increasing the likelihood of the dependent variables showing statistically-significant differences when compared with placebo,our local Institutional Review Board (IRB) strongly insisted against the use of larger doses of levodopa. They argued that participants were healthy volunteers, not patients suffering from dopamine deficiency (i.e., not PD patients), and were seriously concerned that larger doses would lead to an unacceptable higher risk of adverse events. Amongst the untoward effects cited by the IRB where: dangerous decreases in blood pressure, intense nausea and vomiting, and prominent general discomfort. In addition, increasing the levodopa dose could also induce negative effects on cognition due to the inverted U-shaped effects of dopamine (Chowdury et al., 2012). Thus, the final levodopa dose use was decided upon these ethical concerns and the binding request on the part of the IRB.

Regarding risperidone, we also hypothesize that a higher dose would have probably led to significant results vs. placebo. However, these effects would have been confounded by the well-known sedative effects risperidone can induce in healthy volunteers when administered at higher doses.

We have extended the Materials and methods section to better explain the rationale behind our analyses:

“The dopaminergic system has a physiological or intrinsic state whose effects are most likely reflected by the values of the dependent variables measured during the placebo session. […] Thus, drug doses use were decided upon these ethical concerns and the binding request on the part of our local Institute Review Board.”

“The purpose of this study was to elucidate whether modulation of the dopaminergic system influenced the variables under study (i.e., memory, learning and reward), rather than to assess the capacity of the drugs themselves to block or enhance the natural physiological responses influenced by dopamine. Levodopa and risperidone were chosen to "displace" the baseline physiological system in opposite directions: risperidone to lower the effects of physiological dopamine release and levodopa to enhance dopaminergic neurotransmission. Thus, as the objective was to bring the dopaminergic system away from its intrinsic state (i.e., the placebo session) and in opposite directions, our analyses focused in directly comparing the risperidone and levodopa data against each other by using the placebo session as a baseline (but see Appendix 1 for analyses taking into account the three pharmacological sessions separately).”

To comply with the reviewers’ suggestions, we also performed the following analyses, which were again in line with our interpretation. We have used repeated measures ANOVAs with three levels (Risperidone, Placebo and Levodopa) and paired post-hoc t-tests to re-analyze the data for the learning and memory scores and the subjective ratings. For the learning measures, in addition and taking into account the relationship between the PAS and the percentage of learned and remembered words, we have also calculated 3 x 2 mixed repeated measures ANOVAs with Session (Risperidone, Placebo and Levodopa) as the within and Group (H^+^, H-) as the between group factor. The results of these analyses are in agreement with our previous report: the pharmacological intervention succeeds in modulating (i.e., displacing the dopaminergic baseline physiological system) both reward and memory/learning scores (i.e., a significant effect of Session is found).

Specifically, for the main M+ learning condition, we found a main effect of session for the percentage of words learned on Day 1 [F(2,50)=5.041, p=0.01, partial η2=0.168] and the total number of remembered words on Day 2 [F(2,50)=4.793, p=0.012, partial η2=0.161]. Post-hoc paired t-tests with Holm-Bonferroni correction showed, as expected, a significant difference between the Levodopa and the Risperidone sessions for both learning on Day 1 [t(25)=2.95, p_corr=0.02, d=0.580] and the total number of words remembered on Day 2 [t(25)=2.44, p_corr=0.044, d=0.479]. Comparisons of levodopa against placebo were not significant after the correction for multiple comparisons was applied [Learning on Day 1: t(25)=1.629, p_corr=0.214, d=0.319; Total Number of Remembered Words on Day 2: t(25)=-0.240, p_corr=0.812, d=0.047]. The risperidone vs. placebo comparison reached significance for the total number of remembered words on Day 2 [t(25)=-2.90, p_corr=0.023, d=0.569], while was still not significant for the percentage of learned words on Day 1 [t(25)=-1.63, p_corr=0.214, d=0.328]. As expected, the recognition rate on Day 2 yielded no significant main effect of Session [F(2,50)=1.661, p=0.20, partial η2=0.062].

When dividing the participants into two groups according to their PAS scores, the ANOVA results were also in agreement with our main analyses. A significant Group x Session interaction was found for the total number of remembered words on Day 2 [F(2,50)=7.825, p<0.001, partial η2=0.262], while for the percentage of words learned on Day 1 [F(2,50)=2.786, p=0.073, partial η2=0.112] and the recognition rate on Day 2 [F(2,50)=2.615, p=0.084, partial η2=0.106] the interaction was marginally significant.

Regarding the subjective ratings, as expected, there was a significant main effect of Session for Pleasantness [F(2,50)=2.710, p=0.009, partial η2=0.174] and Confidence [F(2,50)=3.70, p=0.032, partial η2=0.129], but not for Arousal [F(2,50)=0.319, p=0.728, partial η2=0.013]. Post-hoc paired t-tests with Holm-Bonferroni correction showed that Pleasantness ratings were higher under Levodopa than under Risperidone [t(25)=2.85, p_corr=0.026, d=0.560], with no significant differences between the levodopa [t(25)=1.584, p_corr=0.137, d=0.311] or risperidone [t(25)=-1.903, p_corr=0.137, d=0.373] against the placebo session. For Confidence ratings no significant differences were found between any pair of sessions after applying the correction for multiple comparisons [Risperidone vs. Placebo: t(25)=-2.275 p_corr=0.095, d=0.446; Risperidone vs. Levodopa: t(25)=2.231, p_corr=0.095, d=0.438; Levodopa vs. Placebo: t(25)=0.521, p_corr=0.607, d=0.102].

We have added a new figure to show the results for the three sessions separately as a figure supplement to Figure 1. See Figure 1—figure supplement 1

Following the reviewers’ advice we have also calculated correlations (using Spearman´s rho) between the PAS and all measures of learning and memory and also all of the subjective ratings at each session (a total of 18 correlations; we used a p<0.05 FDR corrected threshold to control for the multiple correlations calculated). We have created two figures supplemental to Figure 2 to depict these analyses, see Figure 2—figure supplements 1 and 2.

These correlational analyses show that the relationship between the PAS and the different measures of memory and learning and the pleasantness ratings are also modulated by the drug intervention: r scores are consistently lower for risperidone and higher for levodopa sessions.

We think that these additional analyses further show that the pharmacological intervention modulated the reward and memory related measures collected and further provide causal evidence for a dopamine-dependent mechanism instrumental in intrinsically regulated learning.

We have added these three figures as Supplements to Figures 1 and 2, added an Appendix with the complementary analyses of the three sessions separately and updated the Results section accordingly:

“This suggests that the pharmacological intervention was able to modulate measures of reward, memory and online learning selectively during the main M+ condition. Additional analyses using the values for the three sessions separately (instead of the percentage of change from placebo) further confirmed this pattern of results (see the Supplemental Behavioral Analyses section of Appendix 1 and Figure 1—figure supplements 1A and 1C).”

“Additional correlational analyses with the results of each session separately (risperidone, placebo, levodopa) instead of the drug effect only, further confirm a relationship between learning and memory scores and the PAS, with Spearman’s rhos for this relationship being consistently lower than placebo for risperidone and higher than placebo for levodopa (see the Supplemental Correlational Analyses section of Appendix 1 and Figure 2—figure supplement 1). Although the correlation between the drug effect (calculated as the subtraction of the percentage of change from placebo of the levodopa session minus the percentage of change from placebo of the risperidone session) for pleasantness ratings and the PAS was not significant for the main condition of interest (M+, rs=-0.274, p=0.196), raw pleasantness ratings during placebo and levodopa sessions separately were indeed correlated with participants’ individual differences in sensitivity to reward (see the Supplemental Correlational Analyses section of Appendix 1 and Figure 2—figure supplement 2).”

Appendix 1:

“Supplemental Behavioural Analyses

While the purpose of this work was to elucidate whether modulation of the dopaminergic system influenced the variables under study (i.e., intrinsically regulated learning), rather than to assess the capacity of the drugs themselves to significantly block or enhance the natural physiological responses influenced by dopamine, here we provide separated analyses for the three different sessions (risperidone, placebo, levodopa) and the main condition of interest (M+). […] The significant main effects and interactions for both memory and reward measures further demonstrated that: i) the pharmacological intervention attained its main objective, which was to bring the dopaminergic system away from its intrinsic state (i.e., the placebo session) and in opposite directions (see, for example the Pleasantness ratings in Figure 1—figure supplement 1C); ii) that intrinsically regulated learning is a dopamine-dependent mechanism; and iii) that subject-specific reward/hedonic sensitivity drastically alters learning success (modulations for memory and learning scores only happened in highly hedonic individuals, see Figure 1—figure supplement 1B).”

“Supplemental Correlational Analyses

Following the same rationale as with the supplemental behavioral analyses above and taking into account the significant correlations found between the drug effect (calculated as the subtraction of the percentage of change from placebo of the levodopa session minus the percentage of change from placebo of the risperidone session) and the PAS (see Figure 2 in the main text), here we also calculated correlations (using Spearman´s rho) between the PAS and all measures of learning, memory and the subjective ratings at each session separately (a total of 18 correlations; we used a p<0.05 FDR corrected threshold to control for the multiple correlations calculated). […] These results further suggest that: i) the pharmacological intervention was able to "displace" the baseline physiological system in opposite directions, making the learning experience more (levodopa) and less (risperidone) rewarding; ii) that this led to higher and lower learning and retention rates; and iii) more hedonic participants were able to better capitalize in the increase of the “rewarding experience” (see, for example, the high correlation between the PAS and the retention rate on Day 2 in the levodopa session in Figure 2—figure supplement 2).”

3) More details are needed on the PAS scores, including empirical and theoretical evidence for using these measures in the Introduction/Discussion. Second, can the authors comment if any individuals classify as being clinically anhedonic, and if so this should be addressed in the discussion. Third, do PAS scores relate to learning/memory in the placebo condition? Fourth, can the authors provide more details of how they believe anhedonia relates to their learning mechanisms in the discussion.

We thank the reviewers for these suggestions. We have added new information in different parts of the manuscript regarding the rationale and the implications of using the PAS:

Results section:

“Given that our learning task modulates activity within the reward network and is associated with increased subjective reports of pleasure (Ripollés et al., 2014 and 2016), we further tested whether individual differences in sensitivity to reward interacted with the drug intervention to modulate memory benefits (Ferreri et al., 2017; de Vries et al., 2010; Apitz and Bunzeck, 2013). Twenty-four out of the twenty-six participants completed the Physical Anhedonia Scale (PAS, Chapman et al. 1976; mean score = 11.62 ± 5.47). The PAS is a well-validated scale that evaluates difficulty in feeling physical and aesthetic pleasure in response to typical pleasurable stimuli (Padrao et al., 2013; Mas-Herrero et al., 2014).”

Discussion section:

“This, together with the significant correlation between the PAS and the memory and learning scores (see Figure 2), is also in agreement with previous studies showing that anhedonia is associated with both reduced activity and connectivity between regions within the mesolimbic reward pathway (especially between the VS and the SN/VTA; Keller et al., 2013).”

Regarding whether any of the participants were clinically anhedonic there was one participant with a score of 24 that was more than 2 standard deviations above the mean PAS for the whole group. This is the participant that was a bivariate outlier in the correlations between the PAS and the drug effect (levodopa minus risperidone) for the total number of words remembered and also the recognition rate on Day 2 (this was all described in the Results section of the previous version of the manuscript). This participant, being a bivariate outlier, was excluded from those two correlations (note that the correlations were stronger when including the participant) and therefore is not shown in Figure 2. For the sake of clarity in Author response image 1 we depict the correlations with the anhedonic participant included.

**Author response image 1. respfig1:** Correlations between the PAS and the drug effect (calculated as the difference in percentage of change from the placebo session between levodopa and risperidone) with the anhedonic participant (marked with a red circle) included.

This participant has been included in the correlations between the PAS and each learning/memory score on each session shown in Figure 2—figure supplement 1 (see the previous answer 2 of this manuscript) as in those correlations the subject was not a bivariate outlier. This participant exhibits low percentages of learning on Day 1 and especially poor memory measures (only 5 words remembered) on Day 2. We have added more information about this participant in the Results section:

“Importantly, this participant obtained the highest (more anhedonic) score on the PAS (score of 24, more than 2 standard deviations above the mean score of 11.62 of the group).”

The PAS showed a trend for a correlation with the learning and memory scores during the placebo session as shown in the previous answer to point 2 and in Figure 2—figure supplement 1. As stated before, those correlations are now part of Appendix 1 of the new version of the manuscript.

4) Did some individuals perform worse on levodopa than risperidone. If so are these the same individuals across measures of learning, delayed recognition, and affective responses. Addressing this component, in light of the complexity of DA systems, may help connect these findings to the broader literature (i.e., U-shaped DA curves).

The reviewers raise an interesting point here. As can be seen in Figure 2A, there were some participants who performed worse in terms of learning and memory on levodopa than risperidone for the main learning condition (i.e., showed a negative drug effect, calculated as the subtraction of the percentage of change from placebo of the levodopa session minus the percentage of change from placebo of the risperidone session). In particular, the number of participants who performed worse in levodopa than in the risperidone session was 6 for the percentage of learned words on Day 1, 6 for the total number of words remembered on Day 2, 9 for the recognition rate on Day 2 and 9 for the pleasantness ratings. However, there was not a single participant who showed a negative drug effect in all three learning/memory scores and the pleasantness ratings. In contrast, 9 participants showed positive values in all memory and learning scores and in the pleasantness ratings. In this vein, the drug effect for pleasantness did not significantly correlate with any of the drug effects for learning and memory scores (learning on Day 1: r=-0.04, p=0.829; total number of remembered words on Day 2: r=0.108, p=0.608; recognition rate on Day 2: r=0.153, p=0.463).

If we assess the overlap for the learning and memory scores, there was only 1 participant showing a negative drug effect (levodopa minus risperidone) in the percentage of learned words on Day 1, the total number of words remembered on Day 2 and the recognition rate on Day 2 at the same time (while, in contrast, 12 showed positive values on the three measures). Indeed, while the drug effect for the percentage of learned words on Day 1 correlated with the drug effect for the total number of words remembered on Day 2 (r=0.569, p<0.003), the relationship was not significant for the recognition rate on Day 2 (r=-0.084,p=0.689).

Finally, 4 participants did perform poorer in the levodopa than the risperidone session for both the total number of words remembered on Day 2 and the recognition rate on Day 2 (while 15 participants showed positive values in both measures). This last result is not surprising, as these two memory measures are significantly correlated (r=0.638, p<0.001).

Since there was little overlap between these measures we have decided to exclude these analyses from the final version of the manuscript although we are more than willing to include them as supplemental data if the reviewers think that it is of importance.

[Editors' note: further revisions were requested prior to acceptance, as described below.]

As you can see below, there are just a few minor, easily addressable points. These are centered on the following items. First, the authors should qualify the discussion of the results, in particular describing them as fully bidirectional, as a few of the results did not show significant differences across baseline for both drugs. Second, the authors should revise their supplemental across-subject analysis to use each drug separately compared against baseline, as opposed to each correlation on its own. Third, the authors should include the findings (which were detailed in the revision) in the supplement of the number of participants in the L-dopa condition performing worse. Finally, the authors should include additional details of why PAS/anhedonia was used as the measure of reward sensitivity as opposed to other measures.Below you can see the point-by-point responses by each reviewer.Reviewer #1:The authors did a commendable job responding to the revisions in the previous manuscript. I only had a few remaining concerns that should be easily addressable.

We thank the reviewer for all her/his positive comments and for all the suggestions made during this revision, that we believe have greatly improved the quality of our manuscript.

Firstly, the authors throughout the discussion refer to the bi-directionality of the modulation of dopaminergic neurotransmission. Given that some of the findings from the comparisons against placebo were not significant, these claims should be qualified.

We have removed all the claims to the bi-directionality of the dopaminergic modulation throughout the manuscript.

Further, I believe the request for separate across-subject analyses were mis-interpreted in the first round of reviews. Rather, then looking at each correlation in each group separately, I was hoping to see the correlations with scores calculated as Risperidone>Baseline and then separately L-Dopa>Baseline. I believe these will be more compelling correlations as they remove any subject-specific variance.

We thank the reviewer for this suggestion. To comply with the reviewer’s request, we have now calculated the aforementioned analyses. No significant correlations were found between the PAS and the learning and memory scores or subjective ratings calculated as the difference from placebo for the risperidone and levodopa sessions separately (all ps>0.128). There was only a trend for the correlation between the PAS and the percentage of learned words on Day 1 for levodopa minus placebo (r=-0.39, p=0.055) and the total number of words remembered on Day 2 (r=-0.39, p=0.076) also for the changes in the levodopa versus the placebo session. We are not very surprised by these results, as we think that the more compelling correlations are those that describe the relationship between the total drug effects of the different measures (calculated as the subtraction of the percentage of change from placebo of the levodopa session minus the percentage of change from placebo of the risperidone session) and the PAS. As indicated in the manuscript, in our study we used two drugs to "displace" the baseline physiological system of each participant in opposite directions: risperidone to lower the effects of physiological dopamine release and levodopa to enhance dopaminergic neurotransmission. Thus, in our opinion, the subtraction of the levodopa and risperidone drug effects compared to placebo is the measure that better captures the modulation induced by our pharmacological intervention.

In addition, we really want to thank the reviewer for her/his previous suggestion that, although apparently misinterpreted by us, has led to what we believe are very interesting results. As shown in Figure 2—figure supplements 1 and 2, the relationship between the PAS and the different measures of memory and learning were also modulated by the drug intervention: *r* scores are consistently lower for risperidone and higher for levodopa sessions. Moreover, the PAS was correlated with the *pleasantness ratings both during the placebo and the levodopa sessions*. We find these last results of special interest, as they show that learning new words was more enjoyable for people who were more hedonic, even under normal circumstances (i.e., the placebo session). The relationship between reward and word learning was the main topic of the thesis dissertation of the co-first and corresponding author of this work (Pablo Ripollés) and he would very much like to include these analyses in the final version of the manuscript.

Taking into account that i) we designed the protocol to compare levodopa against risperidone results and ii) that we have a particular interest in describing the relationship between word learning and reward, we would like to leave the supplemental section and analyses as they are.

Reviewer #2:1) I think the authors could have done a better job addressing rationale for using the PAS and could include more in the discussion about this. There are many measures of reward sensitivity and anhedonia and there is not much discussion of why this measure was included.

The PAS is one of the easiest to administer, well-validated and intensively used tests for measuring reward sensitivity and anhedonia. In addition, it shows convergent results with more modern tests measuring trait anhedonia, as the Mood and Anxiety Symptom Questionnaire Short Form (Keller et al., 2013) or the Temporal Experience of Pleasure Scale (Favrod et al., 2008). Some other psychometric measures of reward sensitivity and anhedonia that are translated and validated into Spanish, such as the Sensitivity to Punishment/ Sensitivity to Reward Questionnaire (Torrubia et al., 2001) or the BIS/BAS (Caseras, Avila and Torrubia, 2003) show poorer α coefficients—less than 0.78 and 0.73, respectively—than the PAS (0.92; Fonseca-Pedrero et al., 2009). Indeed, we have previously used the PAS in previous studies exploring individual differences in anhedonia trait (Padrao et al., 2013, Mas-Herrero et al., 2014, Martinez-Molina et al., 2016, Mas-Herrero et al., 2018). We honestly think that a discussion regarding the appropriateness of the different tests that measure anhedonia is out of the scope of this work. We want to remind the reviewer that the use of the PAS is discussed both in the Results and the Discussion section of the previous version of the manuscript. We have now further stressed our points regarding the PAS in the Material and Methods section:

“The PAS is one of the most easy to administer and well-validated tests for measuring reward sensitivity and anhedonia. In addition, it shows convergent results with more modern tests measuring trait anhedonia, as the Mood and Anxiety Symptom Questionnaire Short Form (Keller et al., 2013) or the Temporal Experience of Pleasure Scale (Favrod et al., 2008). Some other psychometric measures of reward sensitivity and anhedonia that are translated and validated into Spanish, such as the Sensitivity to Punishment/ Sensitivity to Reward Questionnaire (Torrubia et al., 2001) or the BIS/BAS (Caseras, Avila and Torrubia, 2003) show poorer α coefficients—less than 0.78 and 0.73, respectively—than the PAS (0.92; Fonseca-Pedrero et al., 2009). Indeed, we have previously used the PAS in previous studies exploring individual differences in anhedonia trait (Padrao et al., 2013, Mas-Herrero et al., 2014, Martinez-Molina et al., 2016, Mas-Herrero et al., 2018). Here, the PAS was used as a proxy to reflect the degree of pleasure taken by individuals when engaging in rewarding behavior (Der-Avakian et al., 2012).”

2) Please include the findings about some people performing worse in L-dopa in the supplement.

We have included this information in the new version of the manuscript, as requested:

Results section:

“This suggests that the pharmacological intervention was able to modulate measures of reward, memory and online learning selectively during the main M+ condition (see the Individual Variability of the Drug Effectsection of Appendix I, for a more in depth description of the individual differences found for the drug effect for each measure).”

Appendix I:

“Individual Variability of the Drug Effect

As can be seen in Figure 2A, there were some participants who performed worse in terms of learning and memory on levodopa than risperidone for the main learning condition M+ (i.e., showed a negative drug effect, calculated as the subtraction of the percentage of change from placebo of the levodopa session minus the percentage of change from placebo of the risperidone session). […] Indeed, the drug effects for these two memory measures are significantly correlated (r=0.638, p<0.001).”

3) I apologize for not noticing this before, so this is something for the researchers to consider in the future. There are findings that suggest that individuals who smoke cigarettes who have abstained from smoking have blunted reward anticipation to money (Sweitzer et al., Biological Psychiatry, 2014). In the future I would encourage the authors to perhaps recruit only non-smokers or to include smoking status as a variable in their analyses. I would be curious to know how many participants in their pool were smokers and if their findings differed based on smoking status. I do not think it is necessary to include in the present manuscript.

We thank the reviewer for this interesting suggestion. We will indeed take this into consideration for future studies. Note that participants in our study were excluded if they smoked more than 10 cigarettes a day.

Reviewer #3:The changes made by the authors have sufficiently addressed my concerns. In particular, I feel the added analyses and figures provide a richer presentation of the data (even though many are supplemental) substantially improve the manuscript.

We thank the reviewer for all the comments provided during the revision process which have certainly improved the quality of our manuscript.